# Distinct CD1d docking strategies exhibited by diverse Type II NKT cell receptors

Catarina F. Almeida[1,2,8], Srinivasan Sundararaj[3,8], Jérôme Le Nours[3,4,8], T. Praveena[3,4], Benjamin Cao[5], Satvika Burugupalli[5], Dylan G.M. Smith[5], Onisha Patel[3], Manfred Brigl[6], Daniel G. Pellicci[1,2], Spencer J. Williams [2,5], Adam P. Uldrich[1,2,9]*, Dale I. Godfrey [1,2,9]* & Jamie Rossjohn [3,4,7,9]*

Type I and type II natural killer T (NKT) cells are restricted to the lipid antigen-presenting molecule CD1d. While we have an understanding of the antigen reactivity and function of type I NKT cells, our knowledge of type II NKT cells in health and disease remains unclear. Here we describe a population of type II NKT cells that recognise and respond to the microbial antigen, α-glucuronosyl-diacylglycerol (α-GlcADAG) presented by CD1d, but not the prototypical type I NKT cell agonist, α-galactosylceramide. Surprisingly, the crystal structure of a type II NKT TCR-CD1d-α-GlcADAG complex reveals a CD1d F'-pocket-docking mode that contrasts sharply with the previously determined A'-roof positioning of a sulfatide-reactive type II NKT TCR. Our data also suggest that diverse type II NKT TCRs directed against distinct microbial or mammalian lipid antigens adopt multiple recognition strategies on CD1d, thereby maximising the potential for type II NKT cells to detect different lipid antigens.

[1] Department of Microbiology & Immunology, Peter Doherty Institute for Infection and Immunity, University of Melbourne, Melbourne, VIC 3010, Australia. [2] Australian Research Council Centre of Excellence in Advanced Molecular Imaging, University of Melbourne, Melbourne, VIC 3010, Australia. [3] Infection and Immunity Program and Department of Biochemistry and Molecular Biology, Biomedicine Discovery Institute, Monash University, Clayton, VIC 3800, Australia. [4] Australian Research Council Centre of Excellence in Advanced Molecular Imaging, Monash University, Clayton, VIC 3800, Australia. [5] School of Chemistry and Bio21 Molecular Science and Biotechnology Institute, University of Melbourne, Parkville, VIC 3010, Australia. [6] Department of Pathology, Brigham and Women's Hospital, Harvard Medical School, Boston, MA 02115, USA. [7] Institute of Infection and Immunity, Cardiff University School of Medicine, Heath Park, Cardiff CF14 4XN, UK. [8] These authors contributed equally: Catarina F. Almeida, Srinivasan Sundararaj, Jérôme Le Nours. [9] These authors jointly supervised this work: Adam P. Uldrich, Dale I. Godfrey, Jamie Rossjohn. *email: auldrich@unimelb.edu.au; godfrey@unimelb.edu.au; Jamie. rossjohn@monash.edu

Natural killer T (NKT) cells recognise lipid-based antigens (Ag) presented by the major histocompatibility class I (MHC I)-like molecule CD1d (reviewed in ref. [1]). There are two broad classes of NKT cells, defined as type I and type II NKT cells. The most extensively studied are the type I NKT cells, which express an invariant TCR α-chain (Vα14-J18 in mice, Vα24-Jα18 in humans and strongly respond to the glycolipid α-galactosylceramide (α-GalCer)) (reviewed in ref. [1]). Upon activation, type I NKT cells rapidly produce an array of cytokines and are implicated in a broad range of diseases through their ability to modulate Ag-presenting cells and effector T, B and natural killer (NK) lymphocytes (reviewed in ref. [1]). Type II NKT cells are also CD1d restricted, but they are distinguished from type I NKT cells because they neither express the invariant TCR α-chain that characterises type I NKT cells nor do they recognise α-GalCer (reviewed in refs. [1,2]). The specificity and function of type II NKT cells is poorly understood although some evidence suggests that they play an immunosuppressive role that may oppose the function of type I NKT cells[3,4] (and reviewed in refs. [1,2]). Given that type II NKT cells appear to be more abundant in humans than their type I NKT cell counterparts[5–7] (and reviewed in ref. [8]), it is important to understand the molecular mechanisms underpinning their antigen reactivity and specificity.

The availability of α-GalCer and CD1d–α-GalCer tetramers[9] has facilitated the study of type I NKT cells and the development, functional potential, and molecular basis for Ag-recognition by these cells is now well understood (reviewed in ref. [1]). Conversely, a limited understanding of the Ags recognised by type II NKT cells has hampered the study of these cells, although some studies have identified Ags recognised by some type II NKT cell lines, including sulfatide, β-glucosylceramide (β-GlcCer), phosphatidylglycerol (PG), diphosphatidylglycerol (DPG), lysophosphatidylcholine (LPC) and lysophosphatidylethanolamine (LPE)[10–20]. Structural studies have determined how a type II NKT TCR can bind to sulfatide or lysosulfatide presented by mouse CD1d[21,22], and revealed a docking mode that was distinct to the type I NKT TCR–CD1d–α-GalCer interaction (reviewed in ref. [23]). Namely, type I NKT TCRs bind with a parallel docking mode over the F′-pocket of CD1d, with the TCR α-chain dominating the interaction with the galactose headgroup, while the TCR β-chain binds solely to CD1d (reviewed in ref. [23]). In contrast, the sulfatide-reactive type II NKT TCR docked orthogonally over the A′-pocket of CD1d with only the CDR3β loop interacting with the sulfatide headgroup[21,22]. These studies suggests that the structural and molecular basis for CD1d–Ag recognition between type I and type II NKT cells is fundamentally distinct (reviewed in refs. [1,24]). However, it remains to be determined whether this represents a general dichotomy that distinguishes the mode of Ag recognition by type I and type II NKT cells. We previously identified an atypical population of CD1d-restricted NKT cells that utilise a Vα10Jα50 TCR α-chain that recognise α-GalCer and α-GlcCer. Some of these atypical NKT cells were activated by a mycobacterial Ag, α-glucuronosyl-diacylglycerol (α-GlcADAG)[25] presented by CD1d.

Here, using α-GlcADAG-loaded CD1d tetramers, we characterise a population of type II NKT cells that are selectively reactive to the α-GlcADAG Ag presented by CD1d and do not recognise α-GalCer. Using a panel of synthetic α-GlcADAG analogues, we demonstrate the key moieties of this molecule that facilitate type II NKT TCR recognition. Structural analysis of a type II Vβ8.2+ NKT TCR–CD1d–α-GlcADAG complex revealed a parallel docking mode over the F′-pocket of CD1d, which contrasted the type II NKT TCR–CD1d–sulfatide ternary complexes[21,22]. Extensive CD1d mutational analysis against a panel of diverse self and microbial-antigen-reactive type II NKT TCRs reveal that type II NKT cell TCRs can adopt differing docking

strategies to engage specific CD1d–Ag complexes. Thus, this study defines how a microbial-Ag is recognised by a type II NKT TCR, and demonstrates that type II NKT TCRs can bind with diverse docking modes to CD1d–Ag. This highlights the diverse lipid-based Ag recognition by type II NKT TCRs.

## Results

**Identification of CD1d–α-GlcADAG tetramer+ NKT cells.** We previously characterised a population of α-GlcADAG-reactive NKT cells, but the extent to which NKT cells can recognise this microbial Ag was unknown[25]. To specifically identify α-GlcADAG-reactive NKT cells, NKT cells were first enriched by depletion of immature (CD24+) thymocytes[25], and CD1d–α-GlcADAG tetramers were used to stain mouse thymocytes from wildtype (wt), Jα18−/− and CD1d−/− mice in a BALB/c background (Fig. 1a, Supplementary Fig. 1a). Both CD1d–α-GlcADAG and CD1d–α-GalCer tetramers identified populations from both wt and Jα18−/− BALB/c thymus but not in the CD1d−/− BALB/c thymus (Fig. 1a, Supplementary Fig. 1a), indicating that the populations of tetramer+ cells in both cases were CD1d-dependent. As expected, CD1d–α-GalCer tetramers labelled a clear population of T cells in Jα18−/− mice, albeit much less abundant than their counterparts in wt mice[25]. While CD1d–α-GlcADAG tetramer+ cells were less abundant than CD1d–α-GalCer tetramer+ cells, they were still clearly detected in both wt and Jα18−/− BALB/c thymus and there was a high statistical significance when comparing these cells in Jα18−/− versus CD1d−/− thymus (Fig. 1a).

To determine if the NKT cells identified by CD1d–α-GlcADAG tetramers were distinct from CD1d–α-GalCer-reactive cells, BALB/c thymus samples were co-stained with both CD1d–Ag tetramers using different coloured fluorochromes. Although most wt-derived thymocytes identified by CD1d–α-GlcADAG tetramers co-stained with CD1d–α-GalCer tetramers, a subset of these NKT cells did not (Fig. 1b, Supplementary Fig. 1a). This was very clear in Jα18−/− thymus, where 50% of the CD1d–α-GlcADAG tetramer+ cells did not bind the CD1d–α-GalCer tetramer. Similar to CD1d–α-GalCer-reactive type I NKT cells, the CD1d–α-GlcADAG tetramer+ NKT cells included two main subsets, namely CD4+ or CD4−CD8− double negative (DN) (Fig. 1c) although the ratio of these varied between mice and in some instances, CD4−CD8+ cells were also detected. Reminiscent of type I NKT cells, CD1d–α-GlcADAG tetramer+ cells expressed the activation/memory markers CD44 and CD69 (Fig. 1c). Collectively, these data show that CD1d–α-GlcADAG tetramer+ cells include a mixture of type I and type II NKT cells.

**Diverse CD1d–α-GlcADAG tetramer+ NKT TCRs.** We next determined the TCR sequences used by the CD1d–α-GlcADAG tetramer+ cells that were sorted as single cells from both wt and Jα18−/− BALB/c thymi, following tetramer-associated magnetic enrichment (TAME) based on gates depicted in Fig. 1d and Supplementary Fig. 1b. CD1d–α-GalCer+ CD1d–α-GlcADAG tetramer− type I NKT cells from wt mice were also sorted as controls. Single cell TCR α- and TCR β-chain paired analysis was performed using multiplex PCR, as previously described[26] (Supplementary Table 1). CD1d–α-GalCer tetramer+ cells are known to express the canonical Vα14Jα18+ type I NKT TCR α-chain rearrangement[27]. In contrast, approximately half (12 out of 25 paired TCR sequences) of the CD1d–α-GlcADAG tetramer+ sorted cells expressed Vα10Jα50 TCR α-chain rearrangements, similar to the Vα10+ NKT cells present in Jα18−/− mice that we previously described[25]. Interestingly, four CD1d–α-GlcADAG tetramer+ clones from wt BALB/c mice expressed a TCR α-chain

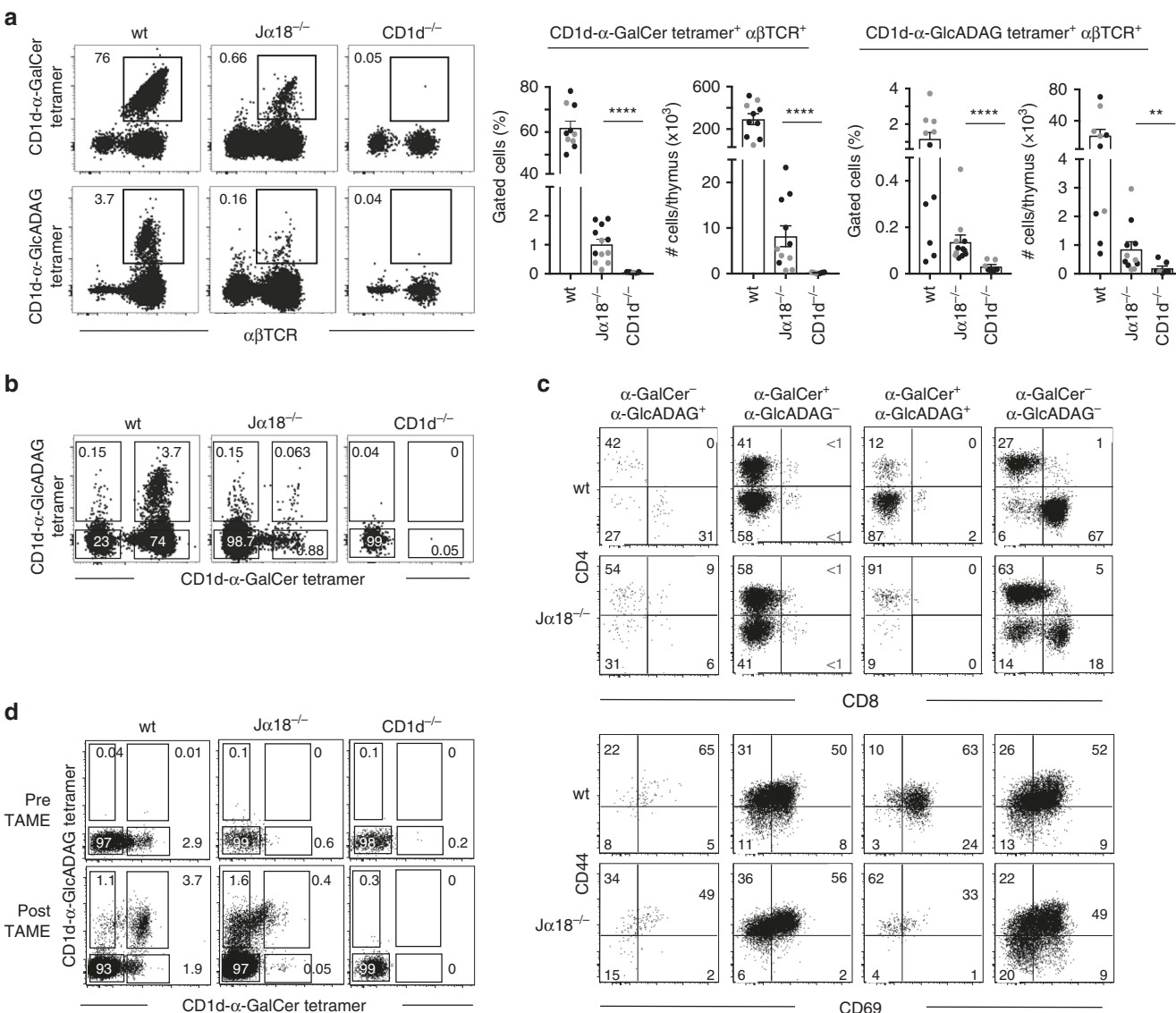

**Fig. 1** Identification of CD1d–α-GlcADAG tetramer+ NKT cells in mice. Flow cytometry analysis of anti-CD24-depleted thymocytes from BALB/c wt, Jα18−/− or CD1d−/− mice **a**. On the left, representative plots showing αβTCR versus CD1d–α-GalCer tetramer (top) or CD1d–α-GlcADAG tetramer staining (bottom) on gated 7AAD−B220−CD11c−CD11b− thymocytes. Numbers next to outlined areas indicate percent cells in each gated population. Graphs on the right show the frequency and total cell numbers per whole thymus, of the population identified by the gate in each plot. Values are representative of a total of $n = 7$ individual experiments where in 5 (out of 7) experiments each black dot represents a pool of five mice (Exps. #1 and #2) or three mice (Exps. #3–5) thymi per group and in 2 (out of 7) experiments the grey dots represent individual thymi. $**p = 0.0079$, $****p < 0.0001$ determined by the two tailed Mann–Whitney $U$ test. **b** Representative plots of dual tetramer labelling of gated BALB/c thymocytes, showing CD1d–α-GlcADAG tetramer versus CD1d–α-GalCer tetramers on 7AAD−B220−CD11c−CD11b−αβTCRint/hi cells. **c** CD4 versus CD8 expression (top), and CD44 versus CD69 (bottom) for each population that has been segregated based on CD1d–α-GlcADAG versus CD1d–α-GalCer tetramer gates in **b**. Plots are derived from four concatenated flow cytometry files acquired in a single experiment, where each file corresponds to a pool of four thymii (representative of two independent experiments). **d** Representative flow cytometry plots showing CD1d–α-GalCer versus CD1d–α-GlcADAG tetramer staining in both pre-enriched and post-enriched samples following CD1d–α-GlcADAG tetramer-associated magnetic enrichment (TAME). Plots depict gated 7AAD−B220−CD11c−CD11b−αβTCRint/hi thymocytes. Numbers indicate percent cells in each gated population. Cells from each population (as identified by gates) were individually sorted into individual wells for TCR gene PCR amplification. In total three independent sorting experiments were performed, where experiments contained a pool of five mice (Exps. #1 and #2) or three mice (Exp. #3), respectively

in which the *Jα50* gene was rearranged with *Vα15*, *Vα4*, *Vα11*, or *Vα4.4*, instead of the *Vα10* gene. These TCR α-chains displayed little or no homology in their CDR1α and CDR2α regions, yet possessed highly similar CDR3α regions suggesting that the Jα50-encoded region confers CD1d–α-GlcADAG recognition in the context of different *Vα* gene usage. This may be due to the conservation of the CDR3α residues Ser109, Ser110 and Phe113 in each of these TCRs, three residues that were involved in the

recognition of CD1d–α-GlcCer complexes by Vα10Jα50+ NKT TCRs[25]. The different CDR1α and CDR2α loops may also facilitate CD1d binding in different ways. Indeed, in a previous study we demonstrated that a Vα10Jα50+ NKT TCR utilised residues within the CDR1 and CDR2 loops to establish contact with CD1d, whilst conserved CDR3α residues contacted both CD1d and the antigen[25]. Interestingly two unique TCR α-chain sequences that did not express *Vα14*, *Jα18* or *Jα50* gene

segments (Vα4Jα17, Vα8Jα49) were also identified amongst wt thymocytes sorted with CD1d–α-GlcADAG tetramers (Supplementary Table 1). TCRβ sequence analysis indicated diversity within the composition and length of the CDR3β regions amongst all sorted populations, and they almost exclusively used Vβ8 (24 out of 25 paired TCR sequences), consistent with predominant Vβ8 gene usage within type I NKT cells[9], Vα10Jα50+ NKT cells[25], and type II NKT cells[28]. Some TCR sequences with identical TCRα and TCRβ nucleotide homology were identified in independently sorted cells (Supplementary Table 1), suggesting clonal expansion in vivo.

Similar results were obtained when NKT cells were isolated from Jα18−/− BALB/c mice (Supplementary Table 1). In the absence of Jα18, Vα10Jα50 TCRs were prominent within the population that stained with CD1d–α-GlcADAG tetramers (5 out of 11 paired TCR sequences), and all used the Vβ8 gene. Five unique TCR α-chain sequences that did not express Vα14, Vα10 or Jα50 gene segments (Vα11Jα9, Vα5Jα12, Vα3Jα4, Vα17Jα2, Vα8Jα16) were observed amongst Jα18−/− thymocytes sorted with CD1d–α-GlcADAG tetramers (Supplementary Table 1). Interestingly, the Vα11Jα9-Vβ8.2 TCR (designated clone name: A11B8.2) was detected in 12 out of 16 sorted clones. As these NKT TCRs displayed non-germline-encoded residues in both the CDR3α and CDR3β regions, this suggests that these T cells had expanded in vivo prior to their isolation.

Overall, the population of thymocytes identified by CD1d–α-GlcADAG tetramers was characterised by heterogeneous TCR usage in both Jα18−/− and wt BALB/c mice, with a range of TCR α-chain sequences and an almost exclusive use of the Vβ8 chain. Moreover, these results demonstrate a means by which microbial Ag-specific type II NKT cells can be isolated.

**Specificity of the CD1d–α-GlcADAG tetramer+ T cells.** A selection of TCR sequences from CD1d–α-GlcADAG tetramer+ NKT cells were used to generate TCR-transduced BW58 cell lines to validate their CD1d–α-GlcADAG reactivity. These included two type II NKT TCRs: Vα11Jα9-Vβ8.2 TCR (clone A11B8.2) and a Vα17Jα2-Vβ8.2 TCR (clone A17B8.2) from Jα18−/− thymus (Supplementary Table 1) and two atypical NKT TCRs: Vα10Jα50-Vβ8.2 TCR and Vα10Jα50-Vβ8.3 (clone A10B8.2 and A10B8.3 from wt thymus (Supplementary Table 1). Controls included BW58 cell lines transduced with: CD1d-sulfatide-reactive XV19 type II NKT TCR[29]; a canonical type I NKT TCR (VB8-STD); and an atypical NKT TCR (Vα10Jα50-Vβ8.1 TCR, clone A10B8.1)[25].

The transduced cell lines were tested for reactivity with CD1d tetramers loaded with a selection of lipid Ags. The Vα10Jα50-Vβ8.2+ and Vα10Jα50-Vβ8.3+ atypical NKT clones (A10B8.2 and A10B8.3) were labelled by CD1d–α-GlcADAG tetramers, but not by CD1d–sulfatide or CD1d tetramers carrying endogenous lipid Ags (CD1d-endo) tetramers, although CD1d–α-GalCer tetramers stained with greater intensity (~25× and 12× higher than α-GlcADAG, respectively, based on tetramer MFI) (Fig. 2a, Supplementary Fig. 1c). Thus, the clones A10B8.2 and A10B8.3 represent examples of atypical NKT TCRs that do not utilise the canonical type I NKT TCR Vα14Jα18 α-chain rearrangement but can still recognise α-GalCer-loaded CD1d[25,30], similar to the control clone A10B8.1[25]. Interestingly, A10B8.1 atypical NKT cells did not label with α-GlcADAG-loaded CD1d tetramers suggesting that the TCR β-chain modulates the differential recognition and Ag-specificity amongst atypical NKT cells, akin to type I NKT cells[31,32].

As expected, the XV19 type II NKT TCR+ cell line showed clear staining with sulfatide-loaded CD1d tetramer whereas the other CD1d tetramers did not stain these cells at higher levels

than CD1d–endo (Fig. 2a). The VB8-STD type I NKT TCR+ cell line was stained by CD1d–α-GlcADAG tetramer with an intensity that was between CD1d–endo tetramer and CD1d–α-GalCer tetramers (Fig. 2a). This shows that some type I NKT cell clones can recognise the microbial Ag α-GlcADAG in the context of CD1d, consistent with the detection of a subset of type I NKT cells by CD1d–α-GlcADAG tetramer (Fig. 1c, d) and the presence of Vα14Jα18 rearrangements amongst cells that co-stained with CD1d–α-GlcADAG and CD1d–α-GalCer tetramers (Supplementary Table 1).

Of particular interest, the type II NKT cell-derived Vα11Jα9-Vβ8.2 TCR+ line (A11B8.2) was clearly labelled by CD1d–α-GlcADAG tetramer (~20× higher than CD1-endo tetramer MFI). In contrast, CD1d–α-GalCer tetramer or CD1d–sulfatide tetramers failed to provide staining of this cell line above CD1d–endo tetramers, highlighting that α-GlcADAG is indeed the preferred ligand for the A11B8.2 TCR (Fig. 2a). The type II NKT TCR+ cell-derived A17B8.2 clone also preferentially bound to α-GlcADAG-loaded CD1d tetramers above CD1d-endo tetramers (~1.5× higher based on tetramer MFI), and interestingly, sulfatide-loaded and α-GalCer-loaded CD1d tetramers displayed a lower ability to stain this line when compared to CD1d-endo tetramers (~2× and 9×, respectively), suggesting that they can act as weaker lipid-antigens for this clone (Fig. 2a).

Collectively, these results highlight the existence of diverse NKT cells expressing TCRs that can interact with CD1d–α-GlcADAG including subsets of type I, type II, and atypical NKT cells.

**CD1d–α-GlcADAG complexes can activate NKT TCR+ cells.** Plate bound CD1d-lipid Ag activation assays were carried out to investigate if the TCR+ cell lines could respond to CD1d–α-GlcADAG recognition (Fig. 2b). The A11B8.2 type II, the A10B8.2 atypical, the A10B8.3 TCR+ atypical, and the VB8-STD type I NKT TCR+ cell lines, were activated by α-GlcADAG-loaded CD1d, leading to CD69 up-regulation and IL-2 production at levels above those elicited by CD1d-endo. Consistent with the tetramer staining (Fig. 2a) the A10B8.2, A10B8.3 atypical NKT TCR+ and the controls A10B8.1 atypical NKT TCR+ and VB8-STD type I NKT TCR+ cell lines responded more strongly to stimulation with CD1d-α-GalCer, while this induced only a weak response from A11B8.2 and A17B8.2 type II NKT TCR+ cells. Notably, and in contrast to A11B8.2, A17B8.2, or VB8-STD NKT TCR lines, no response was detected by the A10B8.1, A10B8.2, and A10B8.3 atypical NKT TCR+ cell line following stimulation with CD1d-endo, implying that these TCRs are highly Ag-dependent (Fig. 2b). Thus, CD1d–α-GlcADAG is capable of stimulating cells expressing both classical type I, type II, and atypical NKT TCRs, in an Ag-dependent manner. This provides the first example of a microbial Ag that can stimulate all three classes of NKT cells.

**TCR–CD1d–Ag-binding affinity.** To understand the specific recognition of CD1d–Ag by type II and atypical NKT TCRs, we measured the binding affinity of soluble forms of the A11B8.2 and A10B8.2 TCRs to CD1d–α-GlcADAG, CD1d–α-GalCer, or CD1d-endo using surface plasmon resonance (SPR). The A10B8.2 atypical NKT TCR bound weakly to CD1d–α-GlcADAG (with a dissociation constant ($K_d$) of ~40 μM), whereas this TCR displayed stronger binding to α-GalCer-loaded CD1d, with a $K_d$ of 11.4 μM, but was not reactive to CD1d-endo ($K_d > 150$ μM) (Fig. 2c), consistent with the lack of CD1d-endo tetramer staining or activation results depicted in Fig. 2a, b. The type II NKT TCR (A11B8.2) bound to CD1d–α-GlcADAG with $K_d$ value of 14.6 μM, which was ~2-fold higher affinity than that observed for

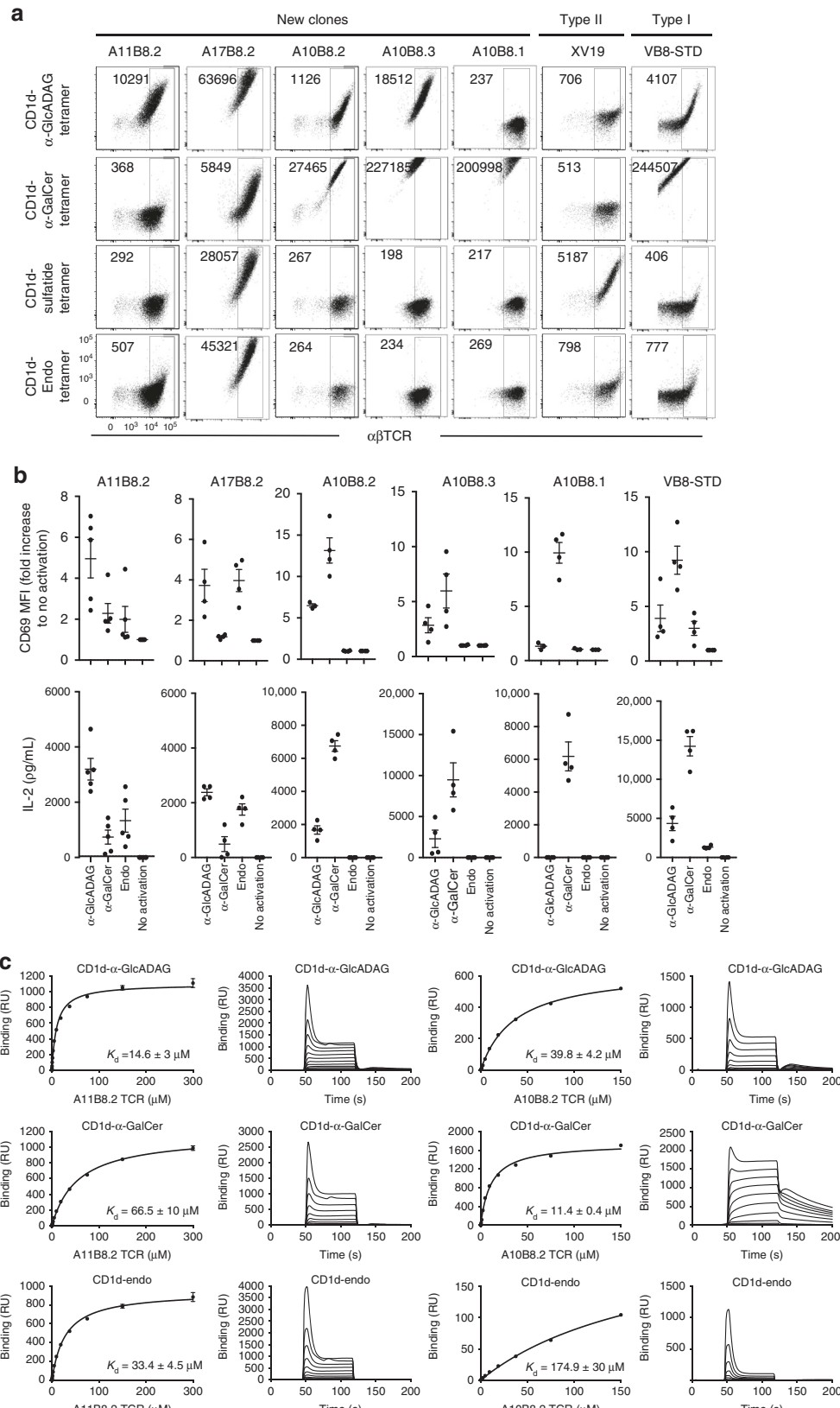

CD1d-endo ($K_d = 33.4\ \mu M$). Binding of this TCR to CD1d–α-GalCer could also be detected, although with a much lower affinity ($K_d = 66.5\ \mu M$) (Fig. 2c). Notably, while binding values can be measured in SPR, differences in binding affinity may reduce the binding threshold that contributes to the lack of tetramer staining detection by flow cytometry. Accordingly, this SPR analysis supports our observations that the A11B8.2 and A10B8.2 TCRs interact with α-GlcADAG presented by CD1d, and moreover, that A11B8.2 preferentially recognises this Ag over the prototypic NKT cell Ag α-GalCer.

**Fig. 2** Antigen reactivity of CD1d–α-GlcADAG tetramer+ sorted clones. **a** α-GlcADAG-, α-GalCer-, sulfatide-loaded or unloaded CD1d (CD1d-Endo) tetramer reactivity of BW58 cells expressing the nominated NKT cell TCRs. Representative plots from $n = 5$ experiments. Numbers depict mean fluorescence intensity (MFI) of CD1d tetramer within the gate encompassing 7AAD⁻ cells with equivalent surface αβTCR levels. **b** CD69 induction and IL-2 production from NKT TCR+ BW58 cell lines shown in **a**, following culture with immobilised CD1d–α-GlcADAG, CD1d–α-GalCer or CD1d–endo for 16 h. Graphs depict the mean CD69 fold difference relative to unstimulated control condition (top), or IL-2 detection in the supernatant (bottom). Error bars depict SEM. When CD1d was used at 10 μg/mL. Data is representative from $n = 4$ ($n = 5$ for A11B8.2) independent experiments. **c** SPR affinity measurement of soluble A11B8.2 and A10B8.2 NKT TCRs to CD1d–endo and CD1d loaded with α-GlcADAG and α-GalCer. The equilibrium curves are representative of one experiment performed in duplicate. Error bars refer to SEM of two replicates. The sensorgrams are representative of one experiment. $K_d$ values are derived from duplicate runs from $n = 3$ (i) and $n = 2$ (ii) independent experiments performed

**Fine specificity of α-GlcADAG-reactive TCRs**. The fine specificity of CD1d–α-GlcADAG-reactive TCRs was examined using CD1d tetramers loaded with a range of synthetic α-GlcADAG variants (Fig. 3a). These included the (R-C19:0/C16:0) analogue bearing R-tuberculostearic acid (R-TBSA) (C19:0) at sn-1 and C16:0 at sn-2 that is expressed by *Mycobacterium smegmatis*[25] (used in Figs. 1–6), along with two synthetic analogues: the regioisomer GlcADAG (R-C16:0/C19:0) bearing R-TBSA at sn-2 and C16:0 at sn-1; and (C18:0/C16:0) that lacks the distinguishing methyl branch of TBSA (Supplementary Fig. 1). In addition, we included a panel of established NKT cell Ags: α-GalCer (C20:2); α-glucuronosylceramide (α-GlcACer C14:0, GSL-1) and α-galacturonosylceramide (α-GalACer C14:0, GSL-1′), both originally from *Sphingomonas* spp.[33,34] and both contain uronic acid head groups, similar to α-GlcADAG; α-glucosyldiacylglycerol (α-GlcDAG C16:0/C18:1 containing vaccenic acid) originally from *Streptococcus pneumoniae*[35]; α-galactosyldiacylglycerol (α-Gal-DAG C17:1/C16:0) originally from *Borrelia burgdorferi*[36]; α-GlcCer (C20:2)[37,38]; the β-linked Ags sulfatide C24:1[15] and GD3 C26:0[39].

The control VB8-STD type I NKT TCR+ cell line and XV19 type II NKT TCR+ cell lines showed the expected patterns of reactivity, with strong binding to CD1d tetramers loaded with α-GalCer for the former, and sulfatide for the latter (Fig. 3a). Moderate binding by CD1d tetramers loaded with α-GlcADAG R-C19:0/C16:0, but not the other variants of this Ag, as well as to other microbial-derived α-linked glycolipids (α-GlcADAG, α-GalACer, α-GlcDAG, and α-GlcCer), was observed for the VB8-STD type I NKT TCR+ cell line, consistent with previous reports[33–36].

For the A11B8.2 type II NKT TCR+ cell line, of the α-GlcADAG variants, only the bona fide mycobacterial species (R-C19:0/C16:0) was clearly bound by the TCR at levels greater than CD1d-endo (Fig. 3a). This type II NKT cell line showed no binding to any of the other CD1d tetramer–Ag complexes, which highlights the fine specificity of this type II NKT cell-derived TCR, and also the importance of both the polar head group and the acyl chain composition and regioselectivity for recognition of this Ag. In contrast, the A10B8.2 atypical NKT TCR+ cell line bound to CD1d tetramers loaded with α-GlcADAG R-C19:0/C16:0 and R-C16:0/C19:0 analogues, but not to the C18:0/C16:0 analogue (Fig. 3a). As expected[25], this atypical NKT cell line stained strongly with both α-GalCer and α-GlcCer-loaded CD1d tetramers and also showed moderate binding to α-GlcDAG from *S. pneumoniae* and both the *Sphingomonas*-derived lipids α-GlcACer and α-GalACer (Fig. 3a). This cell line showed no appreciable binding to tetramers loaded with sulfatide, GD3 or 'endo'.

We investigated more analogues of α-GlcADAG including a C18:1/C16:0 variant that has been isolated from *M. smegmatis* and *Corynebacterium glutamicum*, in which the TBSA is replaced by oleic acid bearing a double bond between C9 and C10[40–43] (Supplementary Fig. 2 and Fig. 3b). This is representative of related compounds that have been isolated from the pathogenic

fungus *Aspergillus fumigatus*[44]. We also synthesised and investigated analogues with R- or S- versions of TBSA and oleic acid-containing α-GlcADAG or α-GlcDAGs, which lack the glucuronic acid on their glycosidic headgroup (Supplementary Fig. 2 and Fig. 3b). The strongest staining of the type II A11B8.2 NKT TCR+ cell line was achieved with CD1d tetramers loaded with the α-GlcADAG variant bearing the R-TBSA chain (R-C19:0/C16:0). Moderate staining of this line was achieved by the oleic-acid (C18:1/C16:0) followed by the S-TBSA variant (S-C19:0/C16:0), and to a much lower degree by the 'isooleoyl' regioisomer α-GlcADAG (C16:0/C18:1) (Fig. 3b). Furthermore, IL-2 was produced in response to plate-bound CD1d loaded with R- and S-TBSA variants of C19:0/C16:0, C18:1/C16:0 or C16:0/C18:1 α-GlcADAGs (Fig. 3b), which demonstrates that these variants can also elicit functional responses of type II NKT cells. Notably, none of the variants lacking the uronic acid on the glycosidic group was able to elicit responses by the A11B8.2 NKT TCR+ cell line nor allow CD1d-tetramer staining, highlighting a high degree of specificity of the A11B8.2 TCR towards the glucuronic acid. These experiments show that the A11B8.2 TCR has the ability to recognise and respond to α-GlcADAG Ags not only from *M. smegmatis* but also other α-glucuronosyl-producing species, such as those in *C. glutamicum* and *A. fumigatus*.

**Overview of the A11B8.2 TCR–CD1d–α-GlcADAG ternary complex**. To gain insight into the mode of recognition of the A11B8.2 TCR for CD1d presenting a microbial α-linked glycolipid Ag; we determined the crystal structure of the unligated TCR and the A11B8.2 TCR–CD1d–α-GlcADAG ternary complex to 1.7 and 3.0 Å resolution, respectively (Supplementary Table 2). The electron density at the A11B8.2 TCR–CD1d interface was unambiguous and unbiased electron density was clearly visible for the headgroup of α-GlcADAG (Supplementary Fig. 3a). The docking strategy adopted by the A11B8.2 NKT TCR was in clear contrast to the orthogonal docking mode of the type II XV19 TCR atop of the CD1d–sulfatide complex (docking angle ~100°) (Fig. 4a–c, Supplementary Fig. 3b)[21,22]. Instead, and reminiscent of the classical type I and atypical NKT TCR–CD1d–Ag ternary complexes (Fig. 4a, b), the A11B8.2 TCR docked parallel over the F′-pocket of the CD1d Ag-binding cleft (docking angle ~15°) (Fig. 4b and Supplementary Fig. 3b). However, unlike the type I NKT TCR–CD1d–Ag ternary complex, and more analogous to the atypical Vα10+ TCR–CD1d–α-GlcCer ternary complex[25], the individual A11B8.2 TCR α- and β-chains was positioned more towards the α2-helix and F′-pocket of CD1d (Supplementary Fig. 3b). Upon A11B8.2 TCR–CD1d–Ag complex formation, the buried surface area (BSA) was ≈1000 Å² (Fig. 4c) similar to the reported BSA of the type I (760–860 Å²) and Vα10 (910 Å²) NKT TCRs. Here, the A11B8.2 TCR α- and β-chains contributed to 57% and 43% of the BSA, respectively. The CDR3α (24% BSA) and CDR3β (24% BSA) loops contributed the most to the TCR A11B8.2–CD1d–Ag interface and to a lesser extent, the CDR1α (13.5% BSA), CDR2α (9% BSA), and CDR2β (9% BSA) loops were also involved in the interaction with CD1d–α-GlcADAG

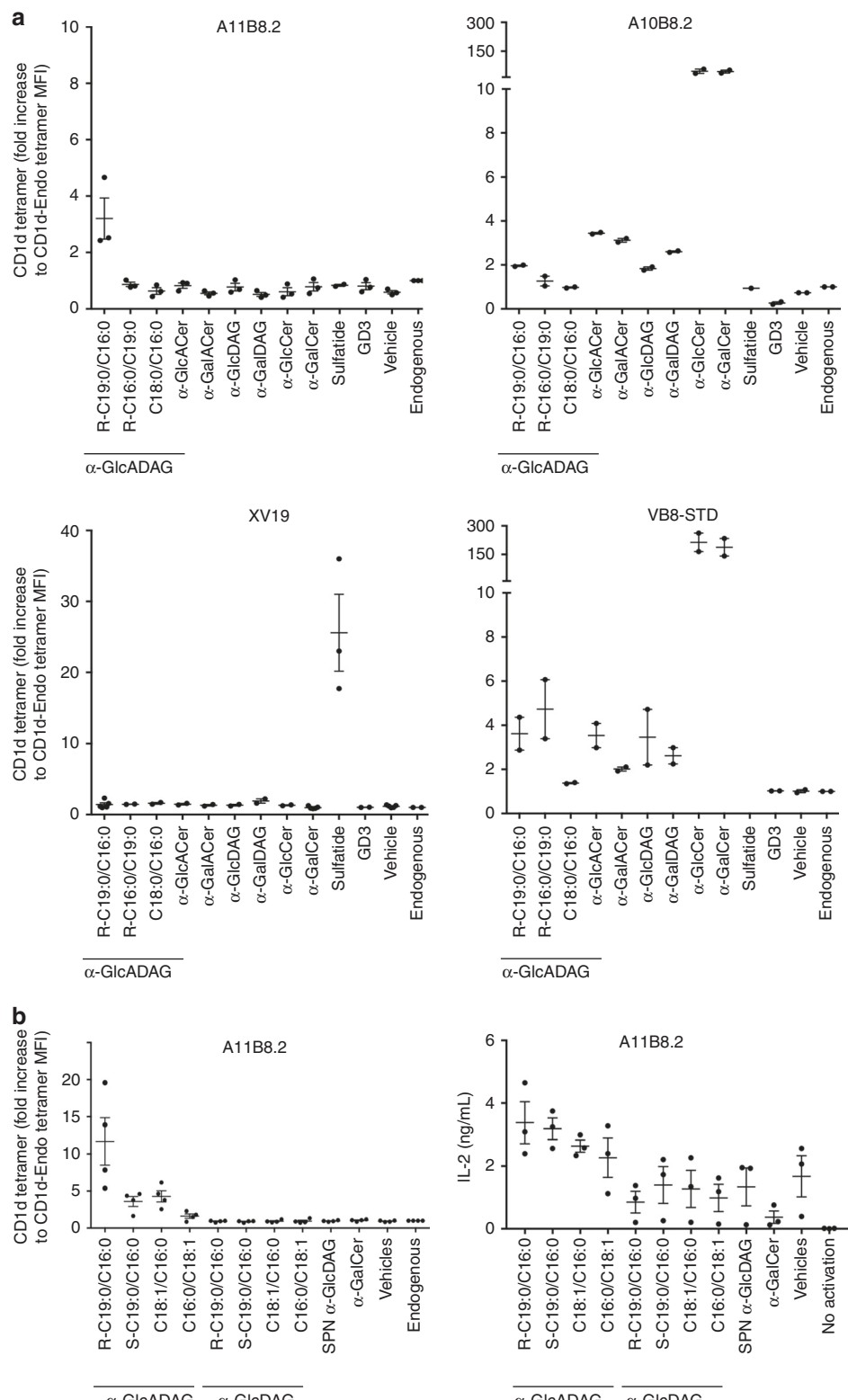

(Supplementary Table 3). Overall, the F′-pocket footprint of the α-GlcADAG-reactive type II NKT TCR ternary complex described here contrasted markedly with the previously determined A′-roof centric docking of the sulfatide-reactive type II NKT TCR.

**Type II NKT TCR/CD1d–α-GlcADAG interactions**. The centrally positioned CDR1α loop mediated contacts with both α1-

and α2-helices of CD1d, whereby Lys31α intersected both helices by forming a salt bridge with Asp80 and van der Waals (vdw) contacts with Asp153 (Fig. 5a), while Thr29α interacted with Val72 of CD1d. The CDR2α loop was positioned directly above the α2-helix of CD1d, in which Ala51α and Gln52α contacted solely Gln154 via vdw and hydrogen bond interactions, respectively (Fig. 5a). The CDR3α loop sat atop the α1-helix of CD1d enabling the aromatic ring of Tyr95α to plunge into a

**Fig. 3** Fine Ag-specificity of BW58 cell lines expressing CD1d–α-GlcADAG-reactive NKT TCRs. **a** BW58 cell lines expressing the A11B8.2 and A10B8.2 TCRs were tested by flow cytometry for their ability to bind a panel of CD1d tetramers loaded with: *Mycobacteria smegmatis* natural occurring α-GlcADAG species (R-C19:0/C16:0) and two synthetic analogue variants (C16:0/C19:0) and (C18:0/C16:0); *Sphingomonas* spp. α-GlcAcer C14:0 (GSL-1); *Sphingomonas* spp. α-GalAcer C14:0 (GSL-1'); *Streptococcus pneumoniae* α-GlcDAG C16:0/C18:1; *Borrelia burgdorferi* α-GalDAG C17:1/C16:0; α-GlcCer C20:2; α-GalCer C:20:2; sulfatide C24:1 and GD3. CD1d-endogenous and vehicle-loaded CD1d tetramers were included as controls. The VB8-STD type I NKT TCR[+] and the XV19 type II NKT TCR[+] cell lines were also included as a control. Graphs represent the mean fluorescence intensity (MFI) of CD1d tetramer staining within cells with similar TCR expression from duplicate values of *n* = 2 (*n* = 3 for A11B8.2) independent experiments and SEM. **b** Synthetic versions of the naturally occurring forms of α-GlcADAG bearing a methyl group with R-enantiomer or S-enantiomer (R-C19:0/C16:0 and S-C19:0/16:0) or the oleic version containing a double bond between C9 and C10 (C18:1/C16:0) or the isooleic variant (C16:0/C18:1), were loaded into CD1d tetramers and assessed for their ability to stain the A11B8.2 type II NKT TCR[+] cell line (left), or to induce IL-2 following 16 h culture on platebound CD1d-lipid (right). Variants in which the glucoronic headgroup (–GlcA) was substituted for a glycosydic headgroup (–Glc) were also tested. *S. pneumoniae* α-GlcDAG (SPN α-GlcDAG), α-GalCer, vehicle-loaded and endogenous-loaded CD1d tetramers were included as controls. Graph on left represents the MFI of CD1d tetramer fold increase over CD1d-endogenous staining within cells with similar TCR expression from four independent experiments ± SEM, and graph on right depicts the mean IL-2 production of three independent experiments ± SEM

hydrophobic pocket lined by Leu84, Val149, and Leu150 (Fig. 5a). Further, Met93α and Gly94α from the CDR3α loop contributed to the interface by contacting Val75, Ser76, Arg79, Asp80, and Glu83 via vdw contacts (Fig. 5a). Within the TCR β-chain, the CDR2β and CDR3β loops dominated the interactions at the interface. Here, the CDR2β loop sat peripherally above the α1-helix of CD1d, whereby Tyr50β contacted the side chains of Met87 and Glu83, while the adjacent framework residues Tyr48β and Glu56β also made contacts with Lys86 and Glu83 or Lys86 and Arg21 (respectively) (Fig. 5b, Supplementary Table 3). Notably, the Vβ8.2 in the type I NKT TCR contains equivalent tyrosine residues in the CDR2β loop (Tyr48β and Tyr50β) that contacted similar residues in CD1d[31]. Further, the CDR3β loop reached across the external side of the CD1d cleft whereby residues Pro96β, Gln97β, Val99β, Ser100β, Tyr101β interacted exclusively with the α2-helix (Leu145, Lys148, Val149, Ala152, and Asp153) (Fig. 5b). The aromatic moiety of Tyr101β pointed towards the CD1d cleft to form hydrogen bonds with the main chain of Val149 and Ala152 (Fig. 5b). Accordingly, the non-germline-encoded CDR3 loops dominated the interactions at the A11B8.2 TCR/CD1d–α-GlcADAG interface (Figs. 4c, 5a, b). This CDR3 loop-mediated dominance at the CD1d–Ag interface was reminiscent of the type II NKT TCR–CD1d–sulfatide complex[21,22] and thus in clear contrast to the type I NKT TCR–CD1d–Ag complex, which is dominated by the germline encoded CDR1α and CDR2β loops and the invariant CDR3α loop (Fig. 4c)[31]. Collectively, the structure of the type II A11B8.2 NKT TCR–CD1d–α-GlcADAG ternary possessed features that were markedly distinct from the type I and type II NKT TCR–CD1d–Ag complexes that were previously characterised structurally.

**Altered headgroup position of α-GlcADAG within CD1d.** The palmitoyl (sn2) and R-TBSA (sn1) chains were positioned within the A′-pocket and F′-pocket, respectively. However, the palmitoyl tail adopted the unusual counter-clockwise orientation that has been previously observed in the type I NKT TCR–CD1d–α-Gal–GSL (PDB code: 3O8X)[45] and CD1d–α-GlcDAG-s2 (PDB code: 3T1F)[35] crystal structures (Fig. 5c). While the overall position of the headgroups of α-GalCer and other microbial-derived Ags (α-Gal–GSL, BbGl2c and α-GlcDAG-s2) generally lays flat in the central region of the Ag-binding groove (Fig. 5c)[23,24,46–48], the α-GlcADAG carbohydrate headgroup clearly leaned markedly towards the A′-pocket of CD1d (Fig. 5d). Here, the glucuronosyl moiety interacted with His68, Met69, Val72, Gly155 of CD1d via vdw contacts, whilst Thr159 formed hydrogen bonds with the 2′- and 3′-hydroxyl (Fig. 5e, Supplementary Table 3). The TCR–α-GlcADAG interactions were exclusively mediated by the TCR α-chain, namely, the

germline-encoded residues within the CDR1α region (Thr29α and Lys31α), and the framework residue Lys67α (Fig. 5f). Here, Lys31α hydrogen bonded to the carbonyl of the stearoyl chain, whilst Thr29α interacted via vdw contacts with the α-GlcADAG carbohydrate moiety. The positioning of the A11B8.2 TCR α-chain indicated that the CDR1α loop would sterically clash with the headgroups of a number of CD1d lipid Ags, including α-GalCer, sulfatide and microbial-derived ligands, consistent with the lack of staining of the A11B8.2 TCR[+] cell line for CD1d tetramers loaded with such Ags (Figs. 2a and 3a). Overall, α-GlcADAG was involved in a coordinated series of interactions, with both CD1d and the A11B8.2 TCR, which allowed it to acquire a unique position where the headgroup of this ligand was tilted towards the A′-pocket of CD1d, which is in contrast to binary and ternary complexes with other CD1d-binding Ags. Thus, the A11B8.2 type II NKT TCR–CD1d–α-GlcADAG ternary complex reveals a distinct mechanism of lipid recognition and provides the first structural insight to diverse mechanisms of CD1d–Ag recognition amongst the type II NKT TCRs.

**Type II NKT TCRs adopt multiple CD1d-binding modes.** To more broadly investigate the docking mode of diverse type II NKT TCRs with CD1d, we generated soluble CD1d molecules with 18 single residue mutations across the length of the Ag-binding cleft and tested their ability to stimulate a panel of NKT cell lines (Fig. 6). These mutants were chosen based on the available CD1d crystal structures, where residues selected were solvent exposed and most did not contact the lipid antigens. These six type II NKT cell lines exhibited diverse TCR usage and reactivity towards self-lipids or microbial lipids, thereby providing a broad perspective on the range of CD1d-docking modes that type II NKT TCRs could potentially employ. As a control, we also examined impact of these mutants on a type I NKT cell line (VB8-STD).

Consistent with previous studies, the type I NKT cell line (VB8-STD) was impacted by the CD1d mutations: Arg79Ala, Glu83Ala, Lys86Ala, Met87Ala, Leu145Ala and Val149Ala. These residues are localised around the F′-pocket and mediate contacts with the type I NKT cell TCRs (Fig. 6)[31]. Further, and consistent with the crystal structure, mutational analyses indicated that Met87, Glu83, Lys86 and Val149 were critical for CD1d–Ag recognition by the A11B8.2 TCR (Fig. 6), and that Leu145, Thr159 and Ala152 also played a moderate role. Thus, similar to type I NKT TCRs, CD1d recognition by the A11B8.2 NKT TCR appears to be highly dependent on CDR2β interactions with the CD1d residues that comprise the "energetic hot spot"[31,49,50]. Similarly, the A17B8.2 TCR[+] cell line also showed an interaction map that correlates with a docking position over the F′-pocket of

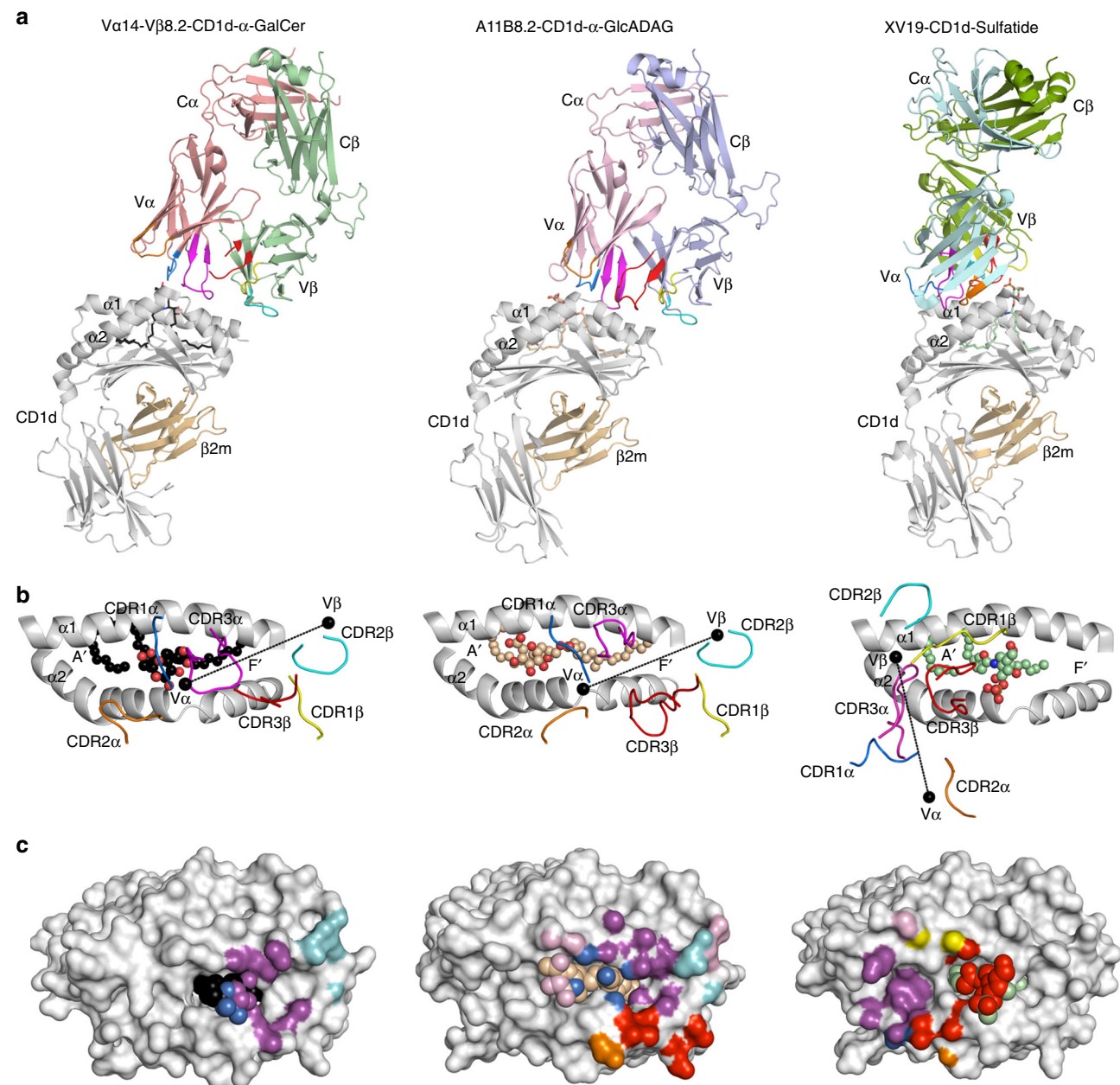

**Fig. 4** Overview of the mouse type I, type II XV19, and A11B8.2 NKT TCR ternary complexes. **a** Cartoon representation of the structure of the mouse type I (Vα14-Vβ8.2) TCR–CD1d–α-GalCer (PDB code: 3HE6) (left panel), mouse A11B8.2 TCR–CD1d–α-GlcADAG (middle panel), and mouse type II XV19 TCR–CD1d–α-GalCer (PDB code: 4EI5) (right panel). The CD1d and β2-microglobulin molecules are coloured in grey and light orange, respectively. Vα14-Vβ8.2 TCRα, salmon; Vα14-Vβ8.2 TCRβ, light green; A11B8.2 TCRα, light pink; A11B8.2 TCRβ, light blue; XV19 TCRβ, green; XV19 TCRα, cyan. The CDR loops are coloured as follows: CDR1α, blue; CDR2α, orange; CDR3α, magenta; CDR1β, yellow; CDR2β, cyan; CDR3β, red. α-GalCer, α-GlcADAG and sulfatide are coloured in black, wheat, pale green sticks, respectively. **b** Top view of the CD1d-binding cleft for each ternary complex. The lipids are shown as spheres and the centre of mass of the respective TRAV and TRBV variable domains are shown as black spheres. **c** TCR footprints on the CD1d–Ag molecular surface. The molecular surface of CD1d is coloured in light grey

CD1d, similar to the A11B8.2 type II NKT TCR, with mutations Arg79Ala, Glu83Ala, Lys86Ala, Val149Ala, Thr159Ala causing a major reduction in activation. For the A11B8.2 and A17B8.2 TCRs, it is possible that the Thr159Ala mutation was indirectly effecting binding by impacting on lipid loading as this residue formed H-bonds with the α-GlcADAG headgroup (Fig. 5).

In contrast, for the type II XV19 TCR[+] cell line, the CD1d mutations that impacted on cellular activation were localised to the A′-pocket (His68Ala, Val72Ala, Thr159Ala, Asp166Ala and Lys65Ala, Gln62Ala, Leu170Ala, and Ser76Ala, Met162Ala) (Fig. 6) which correlates with the contact residues identified in the

crystal complex structure of the XV19 TCR–CD1d–sulfatide[21,22]. The CD1d autoreactive 14S.6, TBA7 and VII68 type II NKT TCR[+] cell lines were assessed using CD1d mutants loaded with 'endo' Ags (Fig. 6)[10,20,51,52]. For the 14S.6 TCR[+] cell line, the interaction map suggested that this TCR spanned across both the A′- and F′-pockets, ranging from Gln62 present in the distal A′-pocket α1-helix, to Leu145, present in the distal F′-pocket. The interaction map generated for the TBA7 TCR (Fig. 6) revealed binding to CD1d over the F′-pocket and closer to the α1-helix. Lastly, the interaction map of the VII68 TCR (Fig. 6) suggested that it bound to CD1d over the extreme F′-pocket, interacting

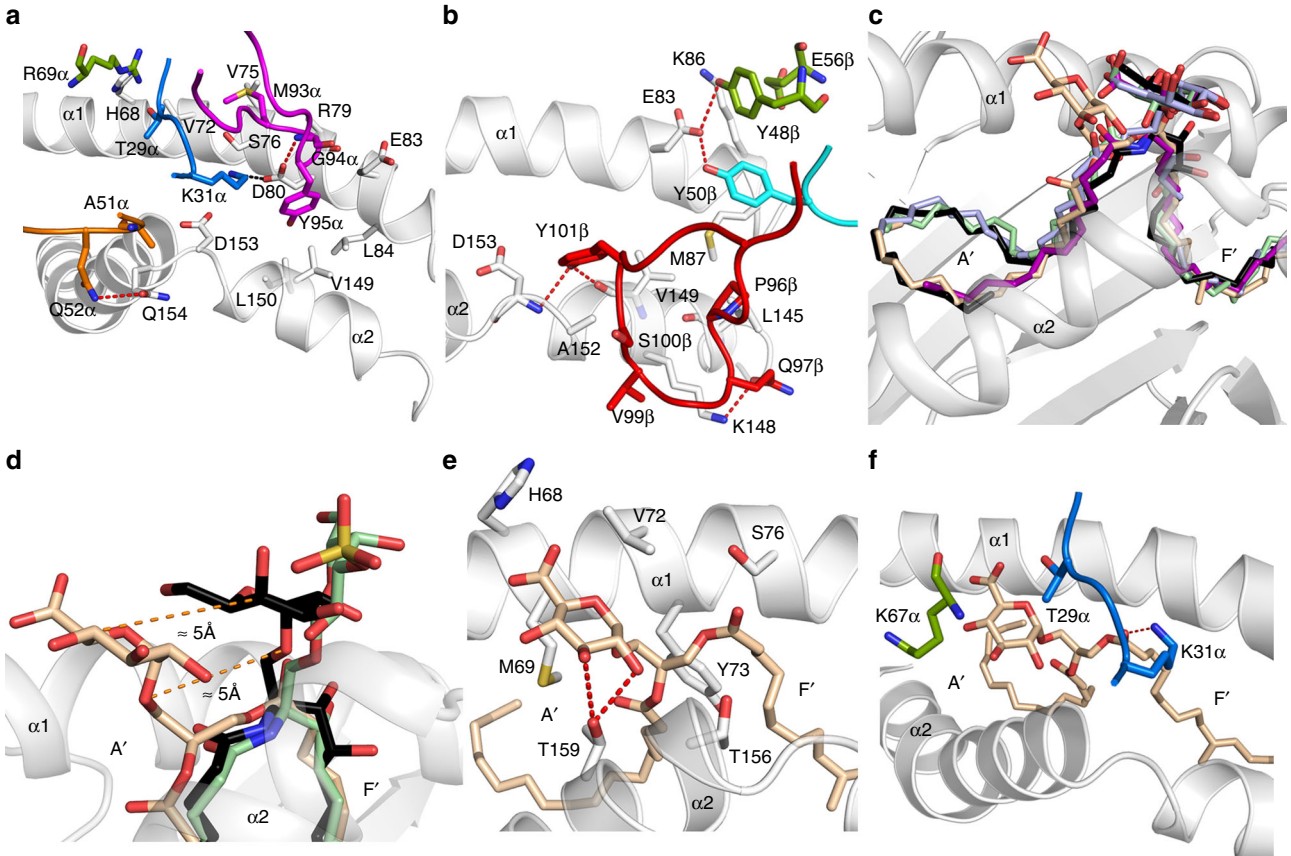

**Fig. 5** Molecular interactions at the A11B8.2 TCR/CD1d–α-GlcADAG interface. **a** A11B8.2 TCR α-chain interactions with CD1d. **b** A11B8.2 TCR β-chain interactions with CD1d. **c** Superposition of NKT TCR–CD1d–microbial lipids ternary structures: CD1d–α-GlcADAG (wheat), CD1d–α-Gal-GSL (purple, PDB code: 3O8X), CD1d–α-GalCer (black, PDB code: 3HE6), CD1d–α-GlcDAG-s2 (light blue, PDB code: 3TA3), and CD1d–BbGL-2c (light green, 3O9W). **d** Superposition of CD1d–α-GlcADAG (wheat), CD1d–α-GalCer (black, PDB code: 3HE6), and CD1d–sulfatide (pale green, PDB code: 4EI5). CD1d is coloured in light grey. **e** CD1d interactions with α-GlcADAG headgroup. **f** A11B8.2 TCR interactions with α-GlcADAG. For clarity, only the hydrogen bonds are shown as red dashed lines and the α1- and α2-helices of CD1d are shown as cartoon representation and coloured in light grey. CDR loops are coloured as in Fig. 5. The framework residues are also coloured in green. CD1d is coloured in light grey

with residues (Glu83, Lys86, Met87 and Val149) located in both α1- and α2-helices.

Taken together, these data suggest that, unlike type I NKT cells, the population of type II NKT cells can adopt diverse docking modes of that span the length of the CD1d-binding cleft. This is likely to reflect the high degree of variability in TCR usage and lipid Ag specificity that is observed for Type II NKT cells.

## Discussion

We have identified a polyclonal NKT cell population that can recognise the microbial glycolipid α-GlcADAG presented by CD1d. To comprehensively confirm the reactivity of this NKT cell population, we have employed a combination of techniques, including single-cell TCR sequencing, TCR-transduced cell line generation and stimulation, and soluble TCR generation for biophysical and structural analysis. The α-GlcADAG used in this study is a synthetic version of one of the most abundant cell membrane lipids in *M. smegmatis*[40]. We also show that these cells can respond to similar α-linked glycolipids including α-GlcAdag from *A. fumigatus* and *C. glutamicum,* α-GlcDAG from *S. pneumoniae* and α-GlcACer from *Sphingomonas* species. The α-GlcADAG-responsive cells include a small subset of classical Vα14Jα18 type I NKT cells and a population of atypical Vα10Jα50 NKT cells. Moreover, we identify here a novel population of α-GlcADAG-specific type II NKT cells. This highlights the

potential for overlap in the Ag specificity between these subclasses of cells and shows that distinct NKT TCRs can adopt diverse molecular mechanisms to interact with CD1d to recognise a common Ag.

While several Ag-specific type II NKT cell populations have been identified[10–20], structural investigations of these TCRs interacting with their CD1d–lipid Ag targets are limited to two studies, both using the same, sulfatide-reactive XV19 TCRs[21,22]. A striking observation from these studies was that the XV19 TCR bound over the A′-roof, in sharp contrast to type I NKT TCRs that invariably bind over the F′-pocket (reviewed in ref. [24]). This raised the question of whether other type II NKT TCRs would adopt this A′-roof-binding mode with CD1d-Ag. We show that type II NKT TCRs can also bind over the F′-pocket of CD1d. Indeed, the structure of the type II A11B8.2 TCR–CD1d–α-GlcADAG complex revealed that the interactions between the two Tyr residues within the CDR2β loop with a defined region over the F′-pocket of CD1d were similar to that observed for mouse Vβ8[+] NKT TCRs[23,31,53]. However, analysis of the binding footprint of a range of type II NKT TCRs with diverse TCR α- and β-chains, indicates that some type II NKT TCRs can bind in multiple ways to CD1d. This likely reflects the high TCR diversity of type II NKT cells and can engender differing reactivities to various CD1d-restricted Ags, including non-lipid molecules[54] or even peptides and lipo-peptides[55]. There are some parallels with that in MR1-mediated immunity, whereby diversity in the

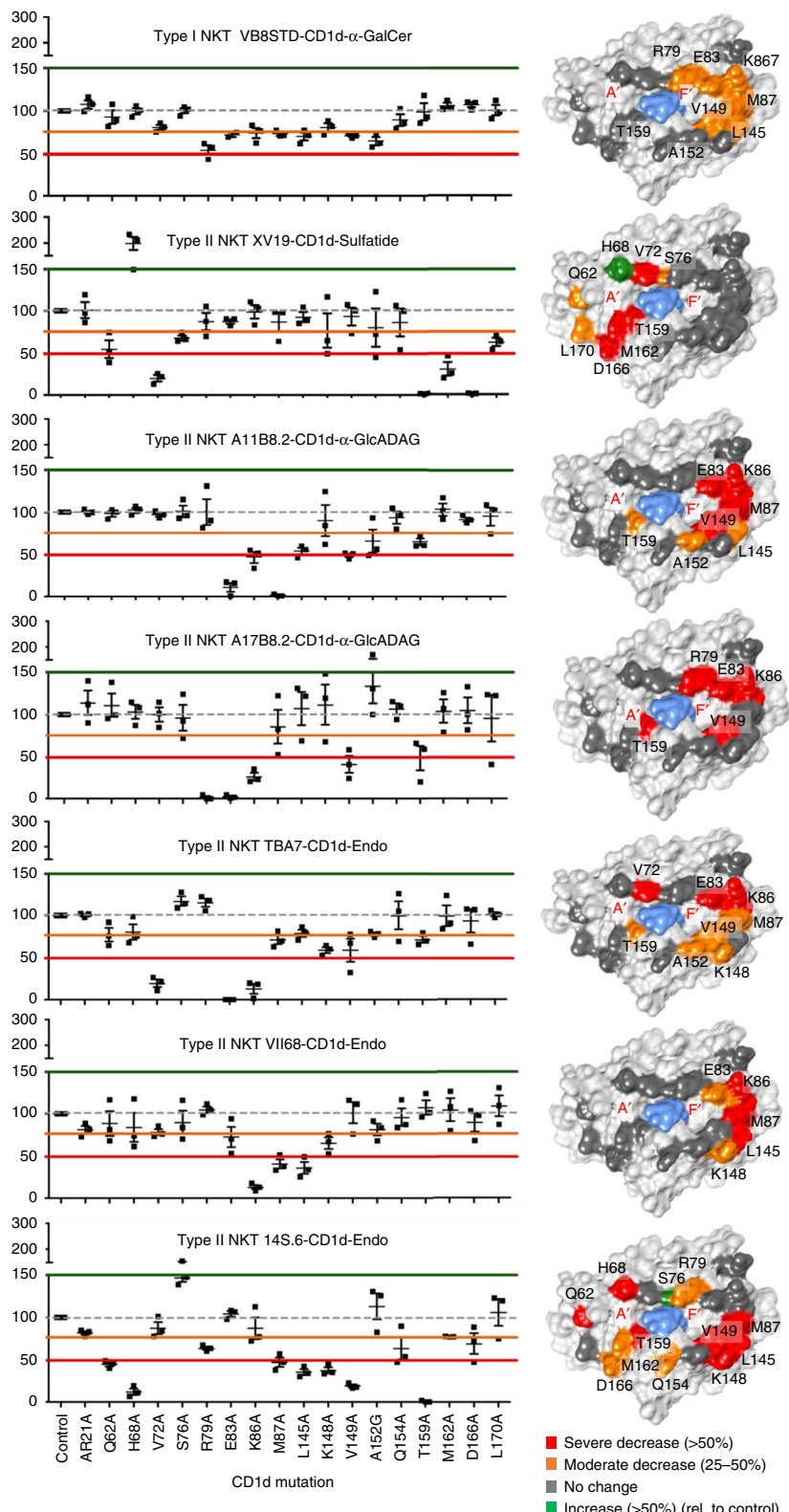

**Fig. 6** Binding modes of a panel of type II NKT cell lines. CD69 up-regulation on NKT TCR-expressing BW58 cell lines, following 16 h culture on plates coated with alanine (Ala) mutants of CD1d loaded with either α-GalCer, sulfatide, or α-GlcADAG. The level of activation elicited by each mutant is normalised to the response elicited by a control CD1d mutant (Asp226Ala). Graphs show the average of duplicate wells ± SEM from one experiment, which is representative of three independent assays. Corresponding surfaces of CD1d (PDB code: 1Z5L) are shown to the right of each graph, depicting residues that when mutated had no effect (dark grey); a 25–50% decrease (orange); a >50% decrease (red); and a >150% increase (green)

MR1-restricted T cell repertoire can also lead to differing reactivities towards metabolite-based ligands and differing docking modes atop MR1[56].

In summary, we identify a new, microbial Ag-specific, population of type II NKT cells. Our findings also suggests that a universal potent Ag for type II NKT cells is unlikely to exist. Rather, our understanding of type II NKT cells will be built upon studies of individual Ag-specific subsets, such as those identified here. Nonetheless, given that type II NKT cells may be more abundant than type I NKT cells in humans[5–7] (and reviewed in ref. [8]), and considering the unique functional roles that these cells appear to play in the immune system (reviewed in ref. [2]) these studies are important if we are to understand the full scope of Ags recognised by type II NKT cells and to harness the therapeutic potential of this arm of the immune system.

## Methods

**Mice**. BALB/c wt, BALB/c Jα18[−/−], BALB/c CD1d[−/−] were bred and maintained according to ethics approval from The University of Melbourne Animal Ethics Committee, under specific pathogen-free conditions at the Biological Research Facility at the Department of Microbiology and Immunology, The University of Melbourne, Australia. BALB/c wt colonies were sourced from either the same facilities, or the Animal Resources Centre (ARC; Canning Vale, WA, Australia). CD1d[−/−] (CD1d1 and CD1d2 double knock-out, described in ref. [57]) mice were sourced from the Peter MacCallum Cancer Centre (Australia) and backcrossed to BALB/c for at least 10 generations. Jα18[−/−] mice were originally provided by M. Taniguchi (Chiba University Graduate School of Medicine, Japan) and backcrossed to BALB/c for at least 10 generations.

**Lymphocyte isolation**. Thymocytes were isolated by grinding thymi through a 50 μM mesh or cell strainer in the presence of 2% FBS PBS. For complement-mediated enrichment of mature thymocytes, the cell suspension was then incubated with anti-CD24 (I) (clone J11d—in house production) on ice for 30 min. Rabbit complement (GTI Diagnostics, Wisconsin, USA) and 70 μg/mL deoxyribonuclease I (DNase; Roche) were then added and allowed to incubate for another 30 min at 37 °C. Thymocytes were washed with 10% FBS/RPMI, then overlayed onto Histopaque-1083 (Sigma), and centrifuged at 1000 × g for 10 min, with no brake, at room temperature. Cells at the interphase were recovered, centrifuged at 350 × g for 4 min at 4 °C and washed with 2% FBS/PBS, or 10% FBS/RPMI for cell culture. All cells were quantified in a Neubauer counting chamber (Merck), with dead cell exclusion by trypan blue solution 0.4% (Sigma).

**Flow cytometry**. All antibodies were acquired from BD Biosciences unless otherwise stated. Fluorochrome-conjugated mouse-specific antibodies include: anti-CD3-allophycocyanin (APC) or fluorescein isothiocyanate (FITC) (clone 145-2C11), anti-TCRβ APC/FITC (clone H57-597), anti-Vβ8.1/8.2 FITC/PE (clone MR5.2), anti-CD4 Brilliant Violet (BV)570/Pacific Blue/Alexa Fluor (AF)700/ BV605 (clone RM4-5), anti-CD8α APC-Cy7 (clone 53-6.7), anti-CD11c FITC/PE-Cy5 (clone HL3), anti-CD11b FITC/PECy5 (clone M1/70), anti-B220 APC-Cy7/ FITC/PE-Cy5 (clone 30-F11), anti-CD69 PE/APC (clone H1.2F3), and anti-CD44 FITC/AF700 (clone IM7). 7-amino-actinomycin D (7AAD) viability dye (Sigma) was included in all flow cytometry-staining panels for dead cell exclusion. Streptavidin (SAV)-PE, SAV-APC, and SAV-BV421 were purchased from BD Biosciences.

Mouse CD1d tetramers were produced in house in High Five insect cells similar to that previously described[9,53,58,59]. In some assays, mouse CD1d was produced in mammalian HEK-293S.N acetylglucosaminyltransferase-I−(GnTI−) (maintained in house > 10 years) cells by cotransfection with pHLsec vectors encoding truncated mouse CD1d ectodomain with a C-terminal biotinylation motif and His6-tag (amino acid sequence at the C-terminus: GSGLNDIFEAQKIEWHEHHHHHH) and β2-microglobulin, using polyethylenimine essentially as described[60]. CD1d was purified from culture supernatant by immobilised metal-affinity chromatography and size-exclusion chromatography, followed by enzymatic biotinylation using biotin ligase (produced in-house), and further purification by size-exclusion chromatography, followed by storage at −80 °C.

Cell suspensions were first incubated for 10 min on ice with Fc-receptor block (anti-CD16/CD32, clone 2.4G2, produced in house), prior to antibody staining. For co-tetramer staining, cells were labelled with the lower affinity CD1d–lipid tetramer first, washed twice and the second CD1d–lipid tetramer (i.e. CD1d–α-GalCer) added together with surface antibodies. For TAME, anti-PE magnetic beads (Miltenyi Biotech) were added to CD1d–α-GlcADAG-labelled cells and the magnetically labelled fraction was isolated using LS columns (Miltenyi Biotech), according to manufacturer's instructions. Cells were analysed on a BD LSRFortessa™ or BD LSR II™, or alternatively, singles cells were sorted on a BD FACSAria III™ (Becton Dickinson) directly into 96-well plates. Samples were analysed using FlowJo software (BD).

**Glycolipids**. α-GalCer (C26:0) was purchased from Alexis Biochemicals. GD3 (C26:0) was purchased from Matreya LLC. Sulfatide (C24:1) was purchased from Avanti Polar Lipids. α-GlcADAG analogues were produced in house. α-GalCer (C20:2), α-GlcCer (C20:2), were supplied by Prof. Gurdyal Besra (University of Birmingham, UK). α-GalCer (C24:1 'PBS-44') and Sphingomonas spp. GSL-1 (α-GlcACer C14:0) and GSL-1′ (α-GalACer C14:0) were provided by Prof. Paul Savage (Brigham Young University, USA). Streptococcus pneumoniae α-GlcDAG (C16:0/C18:1) and Borrelia burgdorferi α-GalDAG (C17:1/C16:0) were provided by Dr. Petr Illarionov (from Gurdyal Besra's Laboratory, University of Birmingham, UK). Glycolipids were prepared in either tyloxapol-based detergent (0.5% or 0.05% v/v tyloxapol in tris-buffered saline (TBS) pH 8, or Tween 20-based detergent (0.5% Tween-20, 56 mg/mL sucrose, 7.5 mg/mL L-histidine in PBS). All glycolipids were stored at −20 °C. Glycolipids were sonicated for ~30 min prior to each use.

**Single cell nested multiplex PCR**. cDNA from individually sorted CD1d tetramer+ αβTCR+ cells was generated by the addition of 2 μl per well of buffer containing SuperScript VILO (Invitrogen) and 0.1% Triton X-100 (Sigma) and incubated according to manufacturer's instructions. cDNA was amplified by two rounds of multiplex nested PCR with a panel of Vα-specific and Vβ primers (Supplementary Table 4)[26,61] and GoTaq Master Mix (Promega). PCR products were separated on a 1.5% agarose gel, purified and sequenced (Applied Genetics Diagnostics, University of Melbourne).

**Generation of TCR-transduced cell lines**. T cell lines expressing αβ TCR were generated by cloning full-length genes encoding α- and β-chains into a pMSCV-IRES-GFP II (pMIG II) vector containing a cis-acting hydrolase element P2A-linked gene system (Addgene plasmid # 52107). The generation of TCR-transduced mouse NKT cell lines was performed similarly to what has been described for the generation of retrogenic mice[62]. Briefly, HEK293T cells (maintained in house > 10 years) were transfected using FuGENE-6 transfection reagent (Promega) with pMIGII expression vector containing both TCR α- and TCR β-chain-verified sequences using vector-specific primers (Supplementary Table 4), pMIGII expression vector containing the CD3 subunits encoding sequences[63], packaging vector pEQ-Pam-3-E and packaging vector pVSV-G[62]. Retrovirus-containing supernatant was collected every 12 h, filtered in a 0.45 μm filter (Sartorius) and used to transduce mouse BW5147.TCR α−β− thymoma cells (termed BW58—maintained in house > 10 years). The pMIG II, expression and packaging vectors were provided by Dr. Dario Vignali (St. Jude's Research Hospital, USA), and the CD3 expression vector was provided by Prof. Stephen Turner (Monash University, Australia). BW58 cell lines were cultured in complete Dulbecco's modified Eagle medium (cDMEM, Gibco) supplemented with 10% FBS (v/v), 15 mM HEPES (Gibco), 1 mM sodium pyruvate (Gibco), 0.1 mM non-essential amino acids (Gibco), 50 μM β-mercaptoethanol (Sigma), 100 U/mL penicillin (Invitrogen), 100 μg/mL streptomycin (Invitrogen) and 2 mM L-glutamine (Invitrogen), at 37 °C, 5% CO2.

For plate bound activation assays, lipid-loaded CD1d was coated in 96-well flat-bottom plates (10 μg/ml) for 3 h at 37 °C. BW58 cell lines were co-cultured with plate bound complex for 16 h in cDMEM. Cells were labelled with CD69 mAb to asses CD69 upregulation (staining panel also included βTCR and 7AAD), and/or culture supernatants were collected for IL-2 cytokine analysis using cytometric bead array (CBA) flex set for mice (BD Biosciences), according to the manufacturer's instructions.

**Soluble TCR generation**. Sequence-verified constructs encoding TCR α- or β-chains in pET30 or pET28 (Supplementary Table 4) were expressed as inclusion body protein (IBP) preparations using Escherichia coli BL-21 (DE3)pLysS and solubilised in 8 M urea, 0.5 mM ethylenediaminetetraacetic acid (EDTA) pH 8.0, 1 mM dithiothreitol (DTT) and 20 mM Tris–HCl pH 8.0. Soluble TCR α- or β-chain IBPs were injected into a refold buffer (containing 0.1 M Tris, 2 mM EDTA, 0.4 M arginine, 0.5 mM oxidised glutathione, 5 mM reduced glutathione, and 5 M urea, and a final pH 8.5). Refolding occurred overnight with gentle stirring at 4 °C. Samples were dialysed for 4 h into 100 mM Urea, 10 mM Tris–HCl pH 8.0 followed by two consecutive dialysis into 10 mM Tris–HCl at pH 8.0 (first for 4 h and second overnight)[31,50]. Refolded TCRs were purified by diethylaminoethanol (DEAE) sepharose anion exchange, followed by Superdex-75 16/60 gel-filtration (GE healthcare) and anion exchange Mono Q 10/100 GL (GE healthcare). The molecular weight of soluble refolded TCRs was validated by liquid chromatography (LC) electrospray ionisation time of flight (ESI-TOF) mass spectrometry (MS) (Agilent, Bio 21 institute).

**Surface plasmon resonance**. SPR experiments were performed on a BIAcore 3000 instrument at 25 °C in HBS buffer (10 mM HEPES, pH 7.4, 150 mM NaCl). 1% BSA was included in the buffer to prevent any non-specific binding. On average, 3000 response units (RUs) of biotinylated-CD1d were coupled onto a SAV (SAV)-coated sensor chip. Biotin was passed over to block-free SAV sites. Serial dilutions of soluble TCRs were passed over as analytes, and experiments were referenced against an empty channel (SAV alone). Data were analysed using Scrubber Pro (BioLogic Software) and BIAevaluation Version 3.1 (Biacore AB). The steady-state dissociation rate ($K_d$; $M$ values) were derived manually from equilibrium analysis in GraphPad Prism 5 using a one site-specific-binding model (1:1 Langmuir kinetic-binding model).

**Crystallisation, structure determination and refinement**. The A11B8.2 TCR (8–10 mg/mL) and A11B8.2–CD1d–α-GlcADAG ternary complex (5–7.5 mg/mL) were crystallised in 10–20% PEG 3350/4% Tascimate, pH 4 and 15–25% PEG 1500/10% succinate phosphate glycine, pH 6.8, respectively, using the hanging drop vapour diffusion method. The crystals were flash frozen and data were collected at the MX1 and MX2 beamlines (Wavelength 0.9537 Å and 100 K) of the Australian Synchrotron, respectively. The data for the A11B8.2 monomer and ternary complex were processed with the XDS software[64] and iMOSFLM 7.0.5/SCALA from the CCP4 suite of programmes[65], respectively. The TCR and the ternary complex X-ray structures were determined by molecular replacement using the software PHASER[66]. For the A11B8.2 TCR, the deposited structures (pdb code: 3QIB for the variable domain Vα11 and pdb code: 3HE6 for the variable domain Vβ8.2) were used as separate search ensembles while for the A11B8.2–CD1d–α-GlcADAG structure, the structures of mouse CD1d minus the antigen (pdb code: 3HE6) and the previously determined A11B8.2 structure were used. The initial experimental phases showed an unbiased density for the headgroup of α-GlcADAG. The programmes BUSTER[67] and COOT[68] were used for refinement and model building, respectively. Subsequent refinement cycles and model building resulted in final $R_{work}/R_{free}$ values of 18.1%/21.9% (For A11B8.2) and 18.7%/24.6% (for A11B8.2–CD1d–α-GlcADAG complex). The quality of both structures was validated at the Research Collaboratory for Structural Bioinformatics Protein Data Bank (RCSB) Validation and Deposition Services. All presentations of molecular graphics were created with the programme PyMOL[69].

**Reporting summary**. Further information on research design is available in the Nature Research Reporting Summary linked to this article.

## Data availability

Atomic coordinates and structure factors of the A11B8.2 NKT TCR and A11B8.2 NKT TCR–CD1d–α-GlcADAG ternary complex were deposited in the Protein Data Bank (PDB) under the accession codes 6MRA [https://www.rcsb.org/structure/6MRA] and 6MSS [https://www.rcsb.org/structure/6MSS], respectively. All remaining data are available within the article and its supplementary information files and from the corresponding authors on request. Source data are provided as a Source Data file.

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

## Acknowledgements

We thank Prof. Michael Brenner for the gift of key type II NKT cell reagents and Maria Sandoval and Christine Wang for technical assistance. We thank the staff at the Australian Synchrotron for assistance with data collection, the staff at the Monash Macromolecular crystallisation facility and the staff from the University of Melbourne flow cytometry facilities. This research was undertaken on the MX1 and MX2 beamlines at the Australian Synchrotron, part of ANSTO. This work was supported by a programme grant from the National Health and Medical Research Council of Australia (NHMRC) (1013667 and 1016629) and the Australian Research Council (ARC) (CE140100011 and DP160100597). C.F.A. was supported by a Fundação para a Ciência e a Tecnologia (FCT) (SFRH/BD/74906/2010); J.L.N. is supported by an ARC Future Fellowship (FT160100074); D.G.P. is supported by a CSL Centenary Fellowship; S.J.W is supported by an ARC Future Fellowship (FT130100103); A.P.U. is supported by an ARC Future Fellowship (FT140100278); D.I.G. is supported by NHMRC Senior Principal Research Fellowship (1117766); J.R. is supported by an Australian ARC Laureate Fellowship.

## Author contributions

C.F.A., S.S. and J.L.N. are joint first authors, performed the experiments, contributed to data generation and analysis and paper writing. T.P., B.C., S.B., D.G.M.S., O.P., M.B., D.G.P., S.J.W. contributed with key reagents and data generation. A.P.U., D.I.G. and J.R. are joint senior and corresponding authors, conceived the study, assisted with data analysis and co-wrote the paper.

## Competing interests

D.I.G. is a member of the scientific advisory board for Avalia Immunotherapies, a company that is developing NKT cell-based vaccines. The other authors declare no competing interests.
