## [Peer Review File · Nature Communications]

Reviewers' comments:

Reviewer #1 (NKT biology, regulatory cells)(Remarks to the Author):

This manuscript has explored the binding specificity of so-called type 2 NKT cells that react with a mycobacterial glycolipid. Prior studies have shown that CD1d molecules can bind a variety of lipid antigens and present them to CD1d-restricted NKT cells that express semi-invariant TCRs (this includes type I NKT cells and a small subset of atypical NKT cells) or oligoclonal/diverse TCRs (called type II NKT cells). Because the different subsets of NKT cells exhibit distinct functions, it is critically important to study their lipid-specificity, as it can inform therapeutic targeting. While much has been learned about type I NKT cells, little is known about type II NKT cells, which are the most abundant NKT cell subset in humans. In the current study, type II NKT cells are identified that react with a mycobacterial lipid (alpha-GlcADAG) bound with CD1d. This interaction is explored by sequencing the TCRs of the type II NKT cells, investigating a variety of lipid variants and CD1d mutants, as well as TCR/CD1d-lipid affinity measurements, and X-ray crystallography to determine TCR/CD1d-lipid interactions. Several similarities but also differences with type I NKT cells reactive with alpha-GalCer, or with type II NKT cells reactive with the endogenous antigen sulfatide, are found. Most interestingly, the type II NKT TCR docks onto the CD1d/alpha-GlcADAG surface (parallel docking) in a manner that is more similar than that of the type I NKT TCR with CD1d-alphaGalCer (parallel docking), rather than another type II NKT TCR with CD1d/sulfatide (orthogonal docking). Overall, the work provides evidence for a surprising amount of diversity in the reactivity and CD1d docking strategy of type II NKT cells.

General comments:

While not extremely innovative, the work provides a detailed and carefully performed analysis of alpha-GlcADAG-reactive type II NKT cells. The work permits comparison with other subsets of alpha-GlcADAG-reactive NKT cells (including type I and atypical NKT cells), as well as type I and type II NKT cells reactive with a variety of other lipids. The work demonstrates a remarkable flexibility in the capacity of distinct NKT cell TCRs to interact with CD1d/glycolipid complexes. Since type I and type II NKT cells exhibit distinct functions, the work is relevant for future therapeutic exploration. It also suggests that type II NKT cells include multiple subsets, with possibly distinct functions.

Minor comments:

1. Figure 1, panels C and D: the numbers in some of the quadrants will need to be aligned better.
2. It is now fairly well recognized that CD1d molecules can also bind peptide or lipopeptide (e.g., Girardi et al. 2016. JBC 291:10677-10683), which might be recognized by type II NKT cells. This additional diversity/flexibility in the recognition system could be mentioned briefly in the discussion.

Reviewer #2 (Structural biology, peptide recognition)(Remarks to the Author):

There are two types of NKT cells – type 1 and type 2. Type 1 are known for having an invariant alpha chain - Va14-Ja18 in mice and Va24-Ja18 in human, and they recognize aGalCer glycolipid. They are also implicated in a range of cytokine production and effector functions. Type II NKT cells do not have a fixed alpha chain and are generally believed to not recognize aGalCer. It is also thought that type II NKT cells immunosuppress type I NKT cell effector functions. Some studies suggest that type II

NKT cells bind to ligands such as sulfatide, bGlcCer, PG, DPG, LPC and LPE. Structural studies have further validated the findings that type II NKTs bind sulfatide or lysosulfatide in complex with CD1d.

Evidence thus far has suggested that the two types of NKT receptors show a distinct binding orientation over their respective binding partners, with type I NKT receptor binding with a parallel binding mode over the F' pocket of CD1d, allowing its alpha chain to extensively interact with the lipid ligand and the beta chain to interact predominantly with the CD1d molecule. In contrast, the NKT II TCR binds with an orthogonal binding mode over the A' binding of the CD1d molecule and only the CDR3b loop interacting with the sulfatide head group. In this paper the authors aim to investigate whether this distinction is a general representation of mode of Ag recognition by type I and type II NKT cells.

The authors identified several clonotypes of type I and type II NKT cells in wild type and Ja18^{-/-} BALB/c mice that appeared specific to CD1d-aGlcADAG tetramer. Interestingly these clonotypes shared little homology in the alpha chain. However they show highly similar CDR3a and a preference for Vb8. The authors further attempted to validate their findings structurally, biochemically, and functionally by characterizing a selected set of Type I and Type II NKT clones. Functional characterization was conducted in the BW58 transduced cell line with CD1d-ligand tetramer sort and cell activation assay, while structural and biophysical characterization was conducted by X-ray crystallography and SPR. The general conclusion of the research suggests that type II NKT TCRs can bind CD1d-ligand in multiple ways, thus resulting in different extents of NKT cell mediated activity against various CD1d epitopes.

These high-level conclusions are favourably supported by the paper. However, to get to these the authors take approaches that need substantial attention. Most importantly, implications of specificity for the class of cells and the receptors they carry are not supported by the data: the flow cytometry and SPR binding data appear to be at odds with each other, and rather loose criteria are used in analyzing the flow data to support the characterization of the cells. In its current incarnation, both sets of data need attention, and implications about specificity in the abstract, introduction, and elsewhere need to be addressed.

Major issues:

1) The subset of the cells which only recognize CD1d-GlcADAG appears rare. However, there are concerns with the flow cytometry data/statistics used to demonstrate this.

* Fig. 1A: Staining of CD1d^{-/-} cells with aGalCer tetramer is 0.05% - this is taken as noise (i.e., not detected as mentioned at the top of the results in Fig. 4). Yet staining of 0.06% in Ja18^{-/-} cells is a signal? The data look like there are more cells in the first set than the second set, but this needs to be addressed, as the distinction between noise (0.05%) and signal (0.06%) appears arbitrary.

* Fig. 1B: Similar issue as above; 0.05% here is taken as an indicator of cells which bound one tetramer but not the other. Is the 0.01% in the WT sample also taken to be an indicator of the existence of these cells, or is it noise?

2. Page 5: "These TCR a-chains displayed little or no homology in their CDR1a and CDR2a regions, yet possessed highly similar CDR3a regions suggesting that the Ja50-encoded region confers CD1d recognition in the context of different Va gene usage." Is it not possible that different CDR1/2 regions could still allow CD1d binding and the conserved CDR3a regions contributing to antigen specificity?

3. Fig. 2: similar issue with regard to flow statistics. If 0.05% is a positive signal in Figs. 1 and 2, what

about the claim that there is no recognition here, e.g. A11B8.2 with GalCer and Endo? Certainly there appears to be stronger recognition with GlcADAG as is claimed so that is fine, but nonetheless quantification and statistics need to be cleaned up.

4. Fig. 2: The SPR data are difficult to interpret and place in context of the story and do not appear to match the flow data. For example, for clone A10B8.2 there is virtually no staining with Endo, yet receptor binding by SPR is clearly shown. The K_d is weaker than for GlcADAG, but by what criteria is a $K_d > 150$ micromolar “nonreactive”, and should that lead to virtually no staining? The situation is worse with the A11B8.2 receptor. There seems to be no staining for anything but GlcADAG, but the K_d differences between GlcADAG, GalCer, and Endo are relatively small – on the order of 2-2.5 fold. Is that presumed to be high specificity, and if so by what criteria? On face value this data does not match the flow data.

The conclusion of this section – that a preference for GlcADAG is in fact supported, but parts of the story elsewhere imply that the cells recognize one but not the other, and the SPR data do not support this. Some of the SPR raw data shown in Fig. 2 look odd: the response values (i.e y axis) in some cases are high, in some cases low (most notably A10B8.2 with GlcADAG and GalCer). Were these different surfaces with the different CD1d injections, or the same? In some cases it is not clear whether a return to baseline was achieved. Perhaps the SPR data are the root of the problem with the discrepancy between flow and binding data?

5. Was glycolipid binding assessed with any of the CD1d mutations? Controls should be presented to ensure that the recognition data observed are not associated with changes in lipid association with CD1d.

Reviewer #1

We thank the reviewer for their positive comments on our paper. We address the minor comments raised below

1. Figure 1, panels C and D: the numbers in some of the quadrants will need to be aligned better.

Thank you for pointing this out. It has been corrected in the revised version.

2. It is now fairly well recognized that CD1d molecules can also bind peptide or lipopeptide (e.g., Girardi et al. 2016. JBC 291:10677-10683), which might be recognized by type II NKT cells. This additional diversity/flexibility in the recognition system could be mentioned briefly in the discussion.

We agree. This reference has now been added in the discussion (page 13).

We now state:

“However, analysis of the binding footprint of a range of type II NKT TCRs with diverse TCR α - and β -chains, indicates that some type II NKT TCRs can bind in multiple ways to CD1d. This likely reflects the high TCR diversity of type II NKT cells and can engender differing reactivities to various CD1d restricted Ags, including non lipid-molecules⁵⁴ or even peptides and lipo-peptides⁵⁵.”

Reviewer #2

We thank the reviewer for their careful evaluation and for stating “These high-level conclusions are favourably supported by the paper.” The reviewer made a number of critiques, which we address in turn below.

1) The subset of cells which only recognize CD1d-GlcADAG appears rare. However, there are concerns with the flow cytometry data/statistics used to demonstrate this.

*** Fig. 1A: Staining of CD1d $-/-$ cells with α GalCer tetramer is 0.05% - this is taken as noise (i.e., not detected as mentioned at the top of the results in Fig. 4). Yet staining of 0.06% in Ja18 $-/-$ cells is a signal? The data look like there are more cells in the first set than the second set, but this needs to be addressed, as the distinction between noise (0.05%) and signal (0.06%) appears arbitrary.**

*** Fig. 1B: Similar issue as above; 0.05% here is taken as an indicator of cells which bound the tetramer but not the other. Is the 0.01% in the WT sample also taken to be an indicator of the existence of these cells, or is it noise?**

The CD1d- α -GlcADAG-reactive cells are less abundant than type-I NKT cells, and our ability to detect them relies on comparing CD1d- α -GlcADAG tetramer staining between CD1d $-/-$ mice (which lack all CD1d-restricted T cells) and J α 18 $-/-$ mice, which lack type-I NKT cells but retain other, less abundant NKT cells. This approach allowed us to detect a population of CD1d- α -GlcADAG tetramer+ cells that were CD1d-dependent, but J α 18-independent. With relatively infrequent cells, it is not ideal to compare tetramers loaded with different lipids (such as α -GalCer) because each tetramer-lipid combination will vary slightly for background staining, just like antibodies do, which can become a problem with less frequent populations. But importantly, we have not simply relied on the tetramer staining because we have carried out other validation studies to confirm the existence of these cells. Furthermore, following Tetramer-Associated-Magnetic Enrichment (TAME), (Fig 1B), CD1d- α -GlcADAG tetramers are able to pull out a population of cells from wt or J α 18 $-/-$ thymi but fail to do so in CD1d $-/-$ thymi cell suspensions, further demonstrating that these are real CD1d-dependent cells.

To further demonstrate the existence of these cells in WT and J α 18 $-/-$ mice, we have carried out additional experiments and added new data and performed statistical analysis of all combined data in the revised manuscript. See updated Figure 1, associated Figure 1 legend and results text on page 4.

A key point of this paper is to demonstrate that the cells identified by CD1d- α -GlcADag tetramers indeed correspond to a population of cells bearing TCRs capable of recognising and responding to CD1d loaded

with this microbial-derived glycolipid antigen. Hence, we have employed a combination of techniques to validate our tetramer-based observations, including TAME, single-cell TCR sequencing, TCR-transduced cell line generation and stimulation, and soluble TCR generation for biophysical and structural analysis. These combined, multi-disciplinary approaches, together with the new data in the revised manuscript, comprehensively validate the existence of these cells.

We now state in the discussion (page 12-13):

“To comprehensively confirm the reactivity of this NKT cell population, we have employed a combination of techniques, including single-cell TCR sequencing, TCR-transduced cell line generation and stimulation, and soluble TCR generation for biophysical and structural analysis.”

2. Page 5: “These TCR a-chains displayed little or no homology in their CDR1 α and CDR2 α regions, yet possessed highly similar CDR3 α regions suggesting that the Ja50-encoded region confers CD1d recognition in the context of different Va gene usage.” Is it not possible that different CDR1/2 regions could still allow CD1b binding and the conserved CDR3 α regions contributing to antigen specificity?

We agree that the CDR1 α and CDR2 α regions may also be contributing to CD1d binding and it is possible that different CDR1 α and CDR2 α residues may facilitate binding in different ways, all in the context of Ja50 as a key component of this interaction. Indeed, in a previous study, we demonstrated how a V α 10+ TCR utilises various residues within the CDR1 and CDR2 loops to establish contact with CD1d, whilst conserved CDR3 α residues contacted both CD1d and the antigen (Uldrich et al, 2011).

We have modified the wording in this section on page 5 to state:

“The different CDR1 α and CDR2 α loops may also facilitate CD1d binding in different ways. Indeed, in a previous study we demonstrated that a V α 10Ja50+ TCR utilised residues within the CDR1 and CDR2 loops to establish contact with CD1d, whilst conserved CDR3 α residues contacted both CD1d and the antigen (Uldrich et al, 2011).”

3. Fig. 2: similar issue with regard to flow statistics. If 0.05% is a positive signal in Figs. 1 and 2, what about the claim that there is no recognition here, e.g. A11B8.2 with GalCer and Endo? Certainly there appears to be stronger recognition with GlcADAG as is claimed so that is fine, but nonetheless quantification and statistics need to be cleaned up.

We are unsure of exactly what the question is here, or what aspect of the quantification and statistics needs to be cleaned up. The key point of Fig 2 was to show that cell lines expressing individual TCRs derived from atypical J α 18- α -GlcADAG-reactive T cells identified in Fig 1 can recognise α -GlcADAG antigens, and that some preferentially bind to this lipid antigen over CD1d-endo or CD1d α -GalCer. This is exemplified by clone A11B8.2, where CD1d-loaded with α -GlcADAG provides much higher tetramer staining (Fig 2A) and stronger activation of cells (Fig 2B). Clone A17B8.2 also recognises α -GlcADAG much more avidly than α -GalCer, but exhibits a strong level of reactivity against CD1d regardless of which antigen was loaded. Clones A10B8.2 and A10B8.3 can also bind to α -GlcADAG (compared to CD1d-Endo tetramer), but not as well as they bind to α -GalCer-loaded CD1d tetramer, whereas clone A10B8.1 shows no reactivity against α -GlcADAG but strongly binds to α -GalCer-loaded tetramer. As these are homogeneous TCR+ cell lines, we are not looking for small percentages of positive staining, like we are in Figure 1. Rather, we are looking for changes to the population of cells. It is also important to consider that TCR-transduced cell lines typically have higher levels of TCR than normal T cells, which can amplify weaker signals, reflecting the low staining for the cells with highest TCR levels as seen for A11B8.2, XV19 and VB8-STD. We hope that this clarifies any concerns here.

To reflect this concern, we now state in the manuscript (page 6)

“In contrast, CD1d- α -GalCer tetramer or CD1d-sulfatide tetramers failed to provide staining of this cell line above CD1d-Endo tetramers, highlighting that α -GlcADAG is indeed the preferred ligand for the A11B8.2 TCR (Figure 2A).

4. Fig. 2: The SPR data are difficult to interpret and place in context of the story and do not appear to match the flow data. For example, for clone A10B8.2 there is virtually no staining with Endo, yet receptor binding by SPR is clearly shown. The K_d is weaker than for GlcADAG, but by what criteria is a $K_d > 150$ micromolar “nonreactive”, and should that lead to virtually no staining? The situation is worse with the A11B8.2 receptor. There seems to be no staining for anything but GlcADAG, but the K_d differences between GlcADAG, GalCer, and Endo are relatively small – on the order of 2-2.5 fold. Is that presumed to be high specificity, and if so by what criteria? On face value this data does not match the flow data.

Tetramer staining intensity and level activation are dependent not only on affinity but also K_{on} and K_{off} rates, as well as avidity which can be influenced by CD1d loading efficiency. Thus, a 2.5-fold difference with SPR can be sufficient to reduce the binding threshold that contributes to the lack of tetramer staining detection by flow cytometry. This has been published by others (Edwards, Lindsay J et al. “Insights into T cell recognition of antigen: significance of two-dimensional kinetic parameters.” *Frontiers in immunology* vol. 3 86. 20 Apr. 2012, doi:10.3389/fimmu.2012.00086).

The conclusion of this section – that a preference for GlcADAG is in fact supported, but parts of the story elsewhere imply that the cells recognize one but not the other, and the SPR data do not support this. Some of the SPR raw data shown in Fig. 2 look odd: the response values (i.e y axis) in some cases are high, in some cases low (most notably A10B8.2 with GlcADAG and GalCer). Were these different surfaces with the different CD1d injections, or the same? In some cases it is not clear whether a return to baseline was achieved. Perhaps the SPR data are the root of the problem with the discrepancy between flow and binding data?

We understand the reviewer’s concern and would like to point out that a lack of return to base line probably indicates a slow off rate. Nonetheless the K_d value was determined in equilibrium for which the lack of return to baseline is not a factor.

To clarify this matter, we now state in the manuscript (page 7):

“Notably, while affinity values can be measured in SPR, differences in binding affinity, on and off rates, and avidity, may all impact on the binding threshold that contributes to the lack of tetramer staining detection by flow cytometry.”

5. Was glycolipid binding assessed with any of the CD1d mutations? Controls should be presented to ensure that the recognition data observed are not associated with changes in lipid association with CD1d.

The CD1d residues selected for mutation were based on the available crystal structures of CD1d. Here, residues were selected that were solvent exposed and most did not contact the lipid antigens. For CD1d mutants showing an impact on activation of VB8-STD, XV19 and A11B8.2 TCRs, we can validate the binding footprints as that they correlate with TCR-CD1d contacts as determined by available ternary crystal structures. Moreover, none of the CD1d mutants chosen had reduced expression levels compared to wild type, indicating that the mutants did not impact on CD1d stability.

We now state in the manuscript (page 11):

“These mutants were chosen based on the available CD1d crystal structures, where residues selected were solvent exposed and most did not contact the lipid antigens.”

The reviewer’s point here may be relevant for T159A mutants in association with α -GlcADAG-reactive TCR clones A11B8.2 and A17B8.2, and we have added the following statement on page 12. “For the A11B8.2

and A17B8.2 TCRs, it is possible that the Thr159Ala mutation was indirectly effecting binding by impacting on lipid loading as this residue formed H-bonds with the α -GlcADAG headgroup (Figure 5).”

In the assays involving cells expressing the TBA7, VII68 and 14S6 TCRs, the CD1d molecules were not loaded with any defined exogenous antigen. As such, while unlikely, we cannot exclude that in these cases CD1d mutations might interfere with the ability of CD1d to bind/position the endogenous lipid antigen headgroup.

Below (Figure 1), we include further mutation impact analysis of CD1d recognition by some of the TCRs in the presence (grey bars) or absence (white bars) of the appropriate exogenously added ligands for these TCRs. The results show a similar pattern of mutations involved in binding, suggesting that the mutations reflect impaired TCR interactions with CD1d, rather than TCR interactions with the lipid antigen. We do not think this figure needs to be included in the manuscript, but include it here for the reviewer’s assessment.

Rebuttal Figure 1.

REVIEWERS' COMMENTS:

Reviewer #1 (Remarks to the Author):

The authors have provided thoughtful responses to the reviewer comments, have provided additional data in support of their findings, and have modified the text accordingly.

Luc Van Kaer

Reviewer #2 (Remarks to the Author):

The authors have responded favourably to my concerns.